

# Rain-on-snow events in mountainous catchments under climate change

Ondrej Hotovy[1], Ondrej Nedelcev[1], Jan Seibert[2], Michal Jenicek[1]

[1]Department of Physical Geography and geoecology, Charles University, Prague, Czechia
[2]Department of Geography, University of Zurich, Zurich, Switzerland

*Correspondence to*: Ondrej Hotovy (hotovyo@natur.cuni.cz)

**Abstract.** The frequency and intensity of rain-on-snow events (RoS) are expected to change in response to climate variations due to changes in precipitation, increase in air temperature and subsequent changes in the snow occurrence. In this study, we attributed these changes to the simulated variations in RoS events using a sensitivity analysis of precipitation and air
temperature, and subsequent effects on RoS-related runoff responses were evaluated. We selected 93 mountainous catchments located in Central Europe across Czechia (60), Switzerland (26) and Germany (7), and used a conceptual hydrological model to simulate runoff components for 24 climate projections relative to the reference period 1980-2010. Climate change-driven RoS changes were highly variable over regions, across elevations, and within the cold season. The warmest projections suggested a decrease in RoS days by about 75 % for the Czech catchments. In contrast, the Swiss
catchments may respond less sensitively, with the number of RoS days even increasing, specifically during the winter months and at higher elevations. Our projections also suggested that the RoS contribution to annual runoff will be considerably reduced from the current 10 % to 2-4 % for the warmest projections in Czechia, and from 18 % to 5-9 % in Switzerland. However, the RoS contribution to runoff may increase in winter months, especially for projections leading to an increase in precipitation, demonstrating the joint importance of air temperature and precipitation for future hydrological
behavior in snow-dominated catchments.

## 1 Introduction

Rain-on-snow (RoS) events threaten society and nature in regions vulnerable to such, often extreme, hydrometeorological events. During RoS events, rain falls on snow and intensifies turbulent, latent, and sensible heat fluxes within the snowpack, which can substantially accelerate snowmelt (Garvelmann et al., 2014; Hotovy and Jenicek, 2020). Although most of these
events do not directly lead to severe flooding, since the snowpack, particularly fresh snow, can store large amounts of rainwater (Juras et al., 2021; Wayand et al., 2015), under certain conditions, these events can also trigger excessive runoff and widespread floods (Berghuijs et al., 2019; Brunner and Fischer, 2022). Elevated runoff generated by RoS is often more intense and short-lived than the thermally driven types of snowmelt and related runoff, along with lower groundwater recharge and infiltration (Earman et al., 2006; Parajka et al., 2019). Thus, such events can affect water supplies and lead to



snow drought. Moreover, RoS events affect important parameters and mechanisms within the snowpack, including changes in snowpack saturation, an increase in liquid water content, and a decrease in snow albedo, which enhances the energy absorption of the snow. These effects can persist for several days after the rainfall event and further accelerate snowmelt (Yang et al., 2023).

The occurrence and intensity of RoS events have been widely studied in recent years, particularly in the Northern

Hemisphere. Although the topic is gaining scientific interest, the complex RoS processes are still on the list of unsolved problems in hydrology proposed by Blöschl et al. (2019).

The most vulnerable regions of the world experience more than 10 RoS events per year (Suriano, 2022). Recent studies have mainly addressed catchments in North America (Bieniek et al., 2018; Crawford et al., 2020; Grenfell and Putkonen, 2008; Musselman et al., 2018), where maximum daily runoff is associated with RoS events mainly (80 % of the time) between

January and May (Il Jeong and Sushama, 2017). Several studies have been conducted in Siberia (Bartsch et al., 2010), Scandinavia (Mooney and Li, 2021; Pall et al., 2019; Poschlod et al., 2020), Central Europe (Freudiger et al., 2014; Hotovy et al., 2023; Juras et al., 2021; Schirmer et al., 2022), high mountain Asia (Maina and Kumar, 2023; Yang et al., 2022), as well as in the terrestrial Arctic (Bartsch et al., 2023). Much of the current research is focused on highlighting the changes in RoS and snow conditions under ongoing climate change.

Despite the increasing scientific interest, future changes in RoS events are still subject to large uncertainties (López-Moreno et al., 2021; Schirmer et al., 2022). The real impact of climate change on RoS events and related hydrologic implications remains unclear, mainly due to their complex nature (Mooney and Li, 2021; Myers et al., 2023; Sezen et al., 2020). This compound effect makes prediction of future RoS changes in complex climate models highly uncertain.

The frequency and intensity of RoS occurrence are expected to change in response to climate variations, including the

distribution, intensity, and phase of precipitation (Blahušiaková et al., 2020; Li et al., 2020; Musselman et al., 2018), as well as the expected increase in air temperature and consequent changes in the snow occurrence (Jennings et al., 2018; Sezen et al., 2020). Snow-related changes will likely become the primary driver of interannual variations in RoS events (Suriano, 2022). Many studies predict a significant decrease in snow storage amounts and duration in the future (Hale et al., 2023; Jenicek et al., 2021; Nedelcev and Jenicek, 2021; Notarnicola, 2020), which is confirmed by observed snow cover duration

(Urban et al., 2023). These changes are expected to be important factors for future RoS occurrences.

Recent studies have also shown that the behavior and occurrence of RoS can be mainly explained by variations in both spatial and temporal distribution. As Hotovy et al. (2023) investigated, various trends in RoS days were identified for specific months of the winter season at different elevations. The largest decrease was observed at lower elevations towards the end of winter, likely due to a shortening of the period with snow cover on the ground. Similar findings were presented by

Beniston and Stoffel (2016); Li et al. (2019); López-Moreno et al. (2021); and Mooney and Li (2021). In contrast, the largest increase was found at higher elevations throughout the winter (Morán-Tejeda et al., 2016; Musselman et al., 2018; Ohba and Kawase, 2020; Sezen et al., 2020; Trubilowicz and Moore, 2017). These changes can be associated with more frequent rainfall during the cold season, triggered by increasing air temperature (Il Jeong and Sushama, 2017; Mooney and Li, 2021).





Although several studies focusing on changes in RoS related to climate change have been carried out, there is still limited
knowledge of the role of different climate variables controlling the RoS behavior and dynamics of the RoS-driven runoff
responses. There is a lack of studies analyzing both changes in RoS and the related runoff responses. Moreover, most
European studies have had a limited focus on elevation, which significantly influences snow cover and precipitation phase
and consequently affects RoS occurrence. Analyzing the combined effect of an increase in temperature and changes in
precipitation is crucial since some studies have shown that the snow decrease caused by the increase in temperature may be
partly offset by the increase in precipitation (Jenicek et al., 2021).

In this study, we present differences between commonly analyzed catchments within the Alpine region and relatively scarce
low-elevation locations outside of this mountain range, representing the areas in the rain-snow transition zones where the
largest changes in snow storage typically occur. Analyzing runoff responses driven by extreme meteorological events within
transition zones is a valuable contribution of this paper, as runoff uncertainty induced by transition elevation is more
pronounced during larger precipitation events (Cui et al., 2023). The detailed temporal and spatial analyses of the effect of
climate change on RoS behavior are also limited. However, understanding these changes and drivers is crucial to future
water management strategies to mitigate risks and impacts associated with RoS events. To address the above research gaps,
the objectives of this study are 1) to attribute changes in selected climate variables to simulated changes in RoS events, using
a sensitivity analysis of precipitation and air temperature, and 2) to evaluate subsequent changes in RoS-related runoff
responses.

## 2 Material and methods

### 2.1 Study catchments

The study included 93 mountainous catchments in two regions within central Europe (Fig. 1). All study catchments with
selected physical and climate characteristics are listed in Table S1 in the Supplementary material. The first regional dataset
(CZ IDs) consists of 60 catchments in six different mountain ranges in Czechia and an additional seven catchments in the
eastern German states of Bavaria and Saxony located within the same cross-border mountain ranges. The original dataset of
40 catchments used in Nedelcev and Jenicek (2021) and Hotovy et al. (2023) was extended by 27 catchments in this study.
The second regional dataset (CH IDs) includes 26 Swiss catchments in three parts of the Alps. For Switzerland, four
catchments were added to the dataset used by Girons Lopez et al. (2020).

These mountainous catchments were selected because they are affected by snow, show near-natural runoff regimes and have
no glacierized areas. Catchment areas range from 1.8 to 478 km$^2$. The mean catchment elevation ranges from 491 to
2434 m a.s.l. The catchments in Czechia and Germany (CZ) generally represent lower elevations than the Swiss catchments
(CH). Annual mean air temperature varies from -0.9 to 8.9 °C. Annual precipitation totals range from 728 to 2187 mm. See
Table S1 for more details at the catchment level.



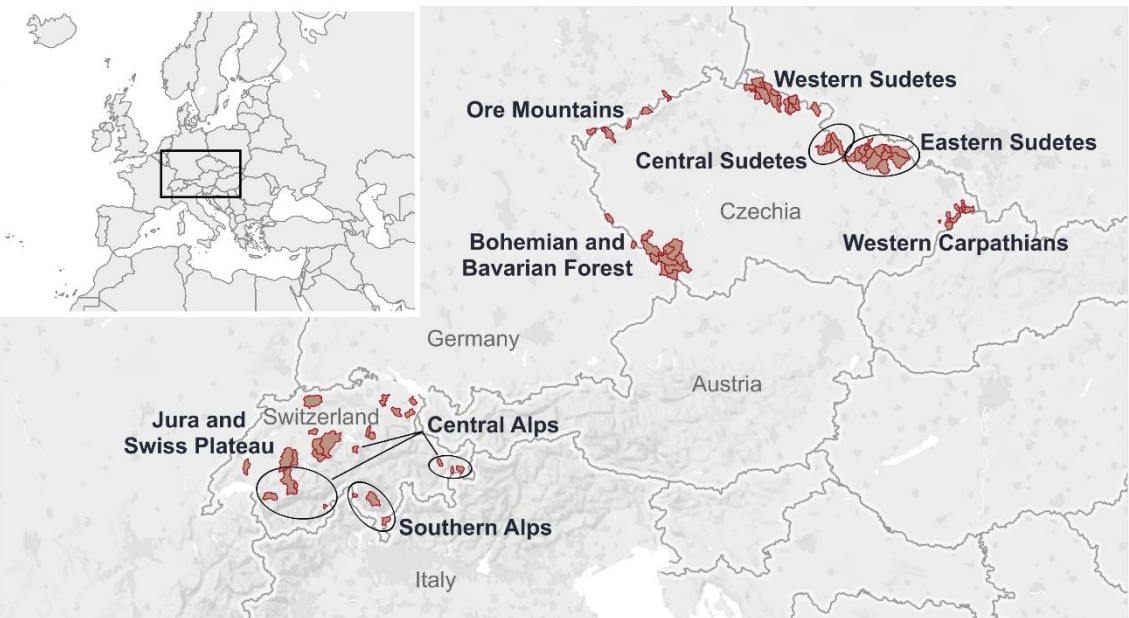

**Figure 1: Location of study catchments in Czechia, Germany, and Switzerland.**

## 2.2 Data

For runoff simulations (Sect. 2.3), a time series of daily mean air temperature, precipitation sums, mean discharge, and weekly snow-water equivalent (SWE) estimates were collected. Stational data for the Czech catchments were available from meteorological and hydrological stations operated by the Czech Hydrometeorological Institute (CHMI). If such a station was unavailable within a given catchment area, the nearest station representing similar conditions and elevations was used. For the German catchments, datasets on temperature and precipitation from the open-source database provided by the German Weather Service (DWD) were used. Discharge data were available from the regional ministries - Landesamt für Umwelt, Landwirtschaft und Geologie (LFULG) for catchments in Saxony (IDs CZ-201, 204-206), and Gewässerkundlicher Dienst Bayern (GDB) for catchments in Bavaria (IDs CZ-101, 102, 112). Temperature and precipitation data for this dataset covers the period 1965-2019. Mean daily discharge and weekly SWE data (taken from the nearest Czech stations) were available for the period 1980-2014.

Data provided by the Swiss Federal Office of Meteorology and Climatology (MeteoSwiss) were used for analyses within the Swiss catchments. The MeteoSwiss gridded data product covers the entire territory of Switzerland and data are available from 1965. We used this data to calculate the mean daily air temperature and precipitation for each catchment. Hydrological data used for the analyses were provided by the Swiss Federal Office for the Environment (FOEN). The mean daily SWE for each catchment was calculated based on a gridded product combining snow depth station data and a snow density model (Magnusson et al., 2014; Mott et al., 2023).





### 2.3 HBV model

To derive individual components of the rainfall-runoff process for the reference period 1980-2010 (30 cold seasons), and to subsequently detect RoS events (Sect. 2.4), a semi-distributed bucket-type HBV model (Lindström et al., 1997; Seibert and Bergström, 2022) in its software implementation "HBV-light" (Seibert and Vis, 2012) was used in this study.

The model is composed of four routines, including a snow routine that simulates snow accumulation and snowmelt using a degree-day approach, taking the potential refreezing of meltwater and snow water holding capacity into account. The

precipitation phase is distinguished by a single threshold temperature ($T_T$) approach, while the $T_T$ values were calibrated separately for each catchment in this study, ranging from -1.66 to 1.16 °C across the Czech catchments and from -1.92 to 1.63 °C for the Swiss catchments. In addition to the snow routine, a soil moisture routine calculates groundwater recharge and actual evapotranspiration (AET) as a function of the soil moisture. For this, the input data of potential evapotranspiration (PET) was calculated based on air temperature data using the method presented by Oudin et al. (2005). Runoff from two

groundwater boxes is simulated by a groundwater routine, from which baseflow is directly calculated by the model. A routing routine calculates the propagation of runoff through the catchment using a triangular function.

Each catchment was split into elevation zones of 100 m. This enables the simulation of some of the characteristics separately for these elevation zones, specifically precipitation, air temperature (using calibrated lapse rates), SWE, snowmelt, soil moisture, AET and groundwater recharge. For details of the model structure and routines, see Seibert and Vis (2012).

The HBV model was calibrated automatically against the observed mean daily runoff and SWE for each study catchment using a genetic algorithm in 100 independent calibration trials. Since the genetic algorithm contains stochastic elements, each calibration trial will result in different optimized parameter sets, especially if there is significant parameter uncertainty (equifinality) (Beven, 2021). Following a split-sample approach, the period was divided into calibration and validation windows for both regional datasets (Table 1). As an objective function, a weighted mean of the NSE (the Nash-Sutcliffe

model efficiency coefficient) based on the logarithmic runoff series (60 %), the volume error (20 %) and the NSE based on the logarithmic SWE series (20 %) was used along with a combination of several objective criteria, which were used for the evaluation of the goodness of fit of the model.

**Table 1: Periods used in the modeling procedure.**

| Model simulation periods | CZ dataset | CH dataset |
|---|---|---|
| Split-sample periods for model calibration and validation | 1981-1997, 1998-2014 | 1981-2000, 2001-2020 |
| Simulated reference period | 1980-2010 | 1980-2010 |


This model setup was similar to the approach used in previous studies, e.g. Seibert and Vis (2012) or Girons Lopez et al. (2020), with various model testing studies carried out, including studies evaluating the overall model performance, e.g. Jenicek and Ledvinka (2020), Nedelcev and Jenicek (2021), or more specific studies assessing the RoS occurrence and SWE, based on model simulations, e.g. Hotovy et al. (2023).



### 2.4 RoS day and RoS event identification

Selection criteria were defined to identify individual RoS days, which happen when rainfall and snow cover occur together. Thus, a RoS day was identified when the following conditions were fulfilled:

1) Precipitation occurred on days with mean temperatures above the threshold temperature $T_T$ (and, thus, being assumed to be rain), with intensities of at least 5 mm per day (i.e., excluding drizzle or negligible amounts). Note that the calibration of the $T_T$ parameter was a part of the general calibration process described in Sect. 2.3.

2) Simulated mean SWE for a given day reaching at least 10 mm, detecting the thick snowpack layer on the ground.

In addition to the RoS day definition mentioned above, multi-day RoS periods, referred to as RoS events here, were identified at a catchment scale to assess hydrological implications and changes in a hydrological response caused by such RoS events. In this study, RoS events were defined as multi-day events, which start from the initial RoS day (the first day when both conditions given above were met), and end day, when the first local maximum runoff was simulated. RoS events may include both, RoS days and non-RoS days. The maximum RoS-driven response time was limited to six days, similar to Freudiger et al. (2014). RoS events were defined in the same way as in Hotovy et al. (2023) and were used for hydrological response analyses (Sects. 2.6 and 3.7).

### 2.5 Sensitivity to climatic variations

A sensitivity analysis assessed how incremental changes in climate variables affect RoS occurrence and their runoff response. In this study, we modified two main climate parameters governing snow storage and RoS events: air temperature (T) and precipitation (P). The modifications consider future changes in these climate variables projected for the central European region by climate models (Gutiérrez et al., 2021). A total of 24 combinations (projections) of increasing air temperature and precipitation change were used for simulations relative to the reference conditions (hereafter referred to as T0_P1). The referenced (current) air temperature (T0) was manually increased by 1-4 °C (T1, T2, T3, T4), and changes in precipitation from a 20 % decrease to a 20 % increase were applied (P08 = -20 %, P09 = -10 %, P11 = +10 %, P12 = +20 %). These 24 combinations cover most of the projected changes in air temperature and precipitation from less warm to warm and from dry to wet conditions. Mean air temperatures and precipitation totals for all catchments in both regions are listed in Table S1 in the Supplementary material. Temperature and precipitation modifications were applied to the entire daily data series.

### 2.6 Assessment of RoS-related variables

As a basis for further analyses, several hydroclimatic statistics (Table 2) were calculated from simulations for each catchment (67+26 catchments) and all 25 projections. These statistics included mean seasonal (Nov-Apr) air temperatures ($T_{mean}$), precipitation sums ($P_{sum}$), annual mean snow water equivalents ($SWE_{mean}$), annual maximum snow water equivalents ($SWE_{max}$), the annual sum of snowfall ($S_{sum}$) and annual snowfall fraction ($S_f$). These annual or seasonal values were then



correlated with the number of RoS days. For this, Spearman's correlation coefficient was used to detect mutual correlations between variables, as the variables were not normally distributed based on the results of a Shapiro-Wilk test. Correlation analyses were performed based on all 25 projections, averaged per catchment, separately for each of the two main geographical regions. Table S2 in the Supplementary material shows modeled values of all variables for distinct projections.

To evaluate the RoS-related hydrological response and its changes for all climate projections, the total runoff ($Q_{\text{event}}$) during RoS events was calculated, and the total direct runoff ($Q_{\text{direct}}$) was calculated for each RoS event (Table 2). A fraction of the total runoff during RoS events to the total runoff was then calculated to describe the relative contribution of the RoS runoff to the total catchment runoff. Moreover, the relative changes in the total direct runoff were evaluated for individual projections.


**Table 2: List of climate and snow variables and hydrological parameters used in the analyses.**

| Variable | Description |
|---|---|
| $T_{\text{mean}}$ | Nov-Apr mean air temperature [°C] |
| $P_{\text{sum}}$ | Nov-Apr precipitation sum (rainfall and snowfall) [mm] |
| $SWE_{\text{mean}}$ | Nov-Apr mean snow water equivalent [mm] |
| $SWE_{\text{max}}$ | Nov-Apr maximum daily/weekly snow water equivalent [mm] |
| $S_{\text{sum}}$ | Nov-Apr snowfall sum [mm] |
| $S_{\text{f}}$ | Nov-Apr snowfall fraction, a ratio of snowfall water equivalent to total precipitation [-]. The threshold temperature calibrated by the HBV model has been used for separating snowfall and rainfall. |
| $Q_{\text{event}}$ | Total runoff during RoS event [mm] |
| $Q_{\text{direct}}$ | Direct runoff (sum) during RoS event [mm] calculated as an outflow from the upper groundwater box of the HBV model, which is considered to be a fast runoff component |

## 3 Results

### 3.1 HBV model evaluation

Overall model performance was evaluated using a combination of selected goodness-of-fit criteria with different weights

(Fig. 2). The median objective function value resulting from 100 parameter sets was 0.76 for model calibration for the Czech catchments and 0.83 for the Swiss catchments (values ranged from 0.56 to 0.86, and 0.70 to 0.87 respectively). Results for model validation reached 0.70 for the Czech catchments, and 0.79 for the Swiss catchments (values ranged from 0.44 to 0.86, and 0.68 to 0.85 respectively). More model testing of SWE simulations and RoS occurrence was carried out by Jenicek et al. (2021), Nedelcev and Jenicek (2021), and Hotovy et al. (2023), who all worked with a similar set of catchments in their

studies. Hotovy et al. (2023) also tested the HBV model performance during RoS events and concluded that the HBV model, despite its conceptualization of the snowmelt process, may be used for RoS analyses, specifically for the assessment of interannual variability and trends of RoS events.




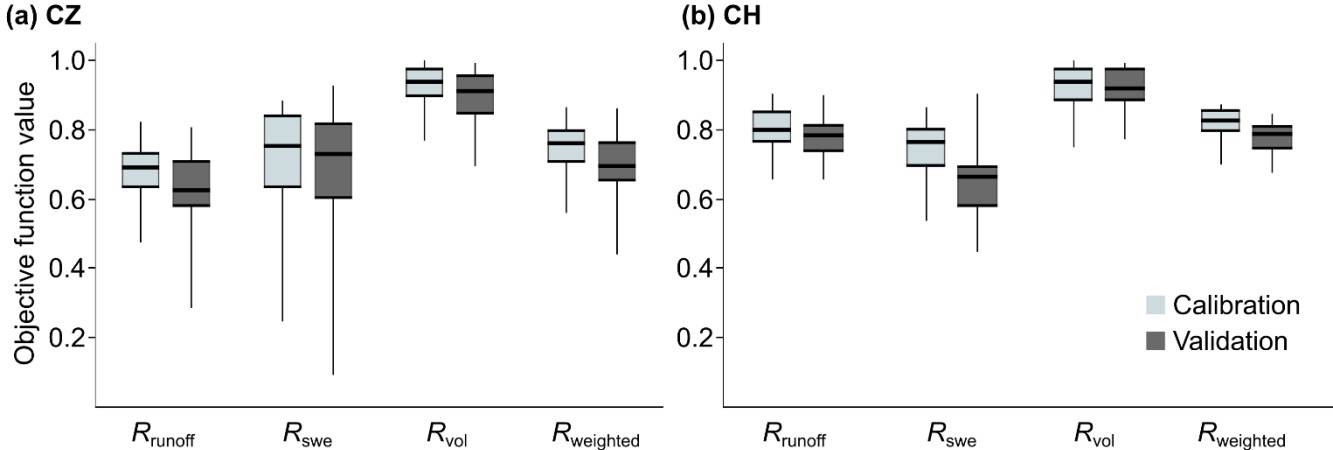


**Figure 2: Model performance for all 93 study catchments within both Czech (a) and Swiss (b) regions evaluated by the combination of selected objective criteria, including the logarithmic Nash-Sutcliffe efficiency for runoff ($R_{runoff}$), Nash-Sutcliffe efficiency for SWE ($R_{swe}$), and volume error ($R_{vol}$). These criteria were weighted ($R_{weighted}$) to calculate the overall objective function of the model. Boxplots represent the variation among catchments, with the 25th and 75th percentiles within a box, the**
**median as a thick line and the whiskers represent maximum and minimum values.**

## 3.2 RoS day occurrence

According to the RoS day definition given in Sect. 2.4, RoS occurrence within individual study catchments and elevations is shown in Fig. 3. The displayed values represent the annual number of RoS days during 30 cold seasons (1980-2010), corresponding to the RoS frequency in the reference scenario T0_P1. Values shown in Fig. 3 are valid for mean elevations of
individual catchments.

The total number of RoS days for each catchment varied from 31 to 1554 in the entire study period. The lowest occurrence was observed at Blanice catchment (CZ-115), where only one RoS day occurred each season on average. In contrast, the Sitter catchment (CH-114) experienced frequent RoS with 52 days each season on average. Generally, the highest number of RoS days appeared within the elevation range of 1000-2000 m a.s.l., including high-elevation Swiss catchments in particular.
At lower elevations, typically for the Czech catchments which experienced the shorter snow season, RoS days occurred less frequently. The number of RoS days decreased at those catchments with the highest mean elevation, likely due to the lack of rainfall during the winter season (results not shown). Distinct catchments saw the average RoS occurrence at different times of the year from mid-January to mid-May, reflecting the increase in elevation.





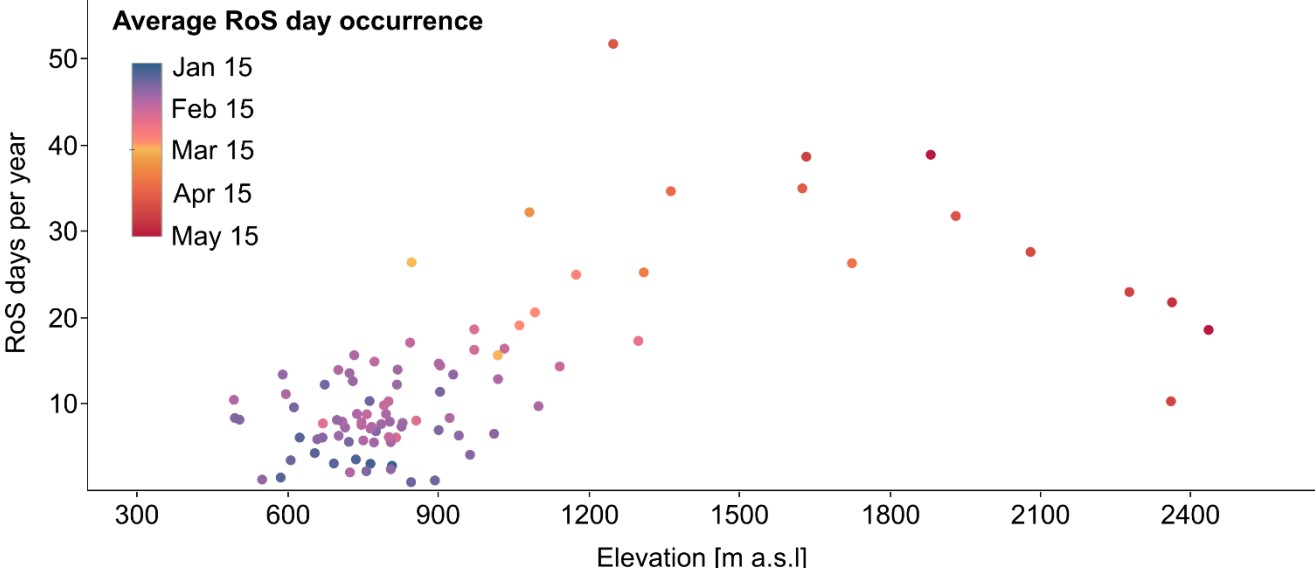

**Figure 3: The annual number of RoS days in all 93 study catchments between 1980 and 2010, corresponding to the reference scenario T0_P1. Each dot represents one catchment and is colored according to the average RoS occurrence throughout the year. Individual catchments are characterized by mean catchment elevation (shown in Table S1).**

### 3.3 Regional changes in RoS days for different climate projections

The numbers of annual RoS days were found to vary regionally and for the 25 projections (Fig. 4). The changes in the median values of all catchments within each region suggest that only four projections for the Czech dataset, and five projections for the Swiss dataset, will lead to an increase in the number of RoS days. The number of RoS days increased only for the projections with a 1 °C temperature increase combined with a precipitation increase (projections T0_P11, T0_P12, T1_P11, T1_P12). In most of the projections, the number of RoS days is expected to decline (Figs. 4a and 4c), especially for projections with a relatively large temperature increase amplified by precipitation decrease. Projections with a temperature increase of 4 °C suggested a decrease of RoS days by about 75 % for the Czech catchments (Fig. 4b). For the high-elevation Swiss catchments, the number of RoS days decreased less (Fig. 4d). However, there were large variations among the individual study catchments in each region.



**Figure 4: Number of RoS days per year in both Czech (a, b) and Swiss (c, d) regions for all projections (a, c), and a fraction of the number of RoS days relative to reference conditions in both regions for all projections (b, d). Boxplots represent the variation among catchments, with the 25th and 75th percentiles represented by each box, the median as a thick line and the whiskers showing the maximum and minimum values. Boxes are grouped and colored according to the temperature (T) and precipitation (P) projections.**

The above-mentioned differences and projected changes across regions are supplemented by the evaluation across individual mountain ranges (Fig. 5). Results showed that catchments are generally less sensitive to changes in precipitation compared to the increase in temperature. This was shown by projections assuming a temperature increase by 4 °C, where similar RoS decreases were suggested among individual mountain ranges, independent of changes in precipitation.

Overall, there were large differences between the individual mountain ranges and selected projections. Regionally, the catchments located in the Western Sudetes will be relatively unaffected by the temperature increase by 2 °C, however, additional temperature rise (T4 projections) may result in sudden RoS decline, which will be more pronounced compared to the Eastern Sudetes and Western Carpathians. In Switzerland, the catchments located in the Central and Southern Alps showed higher resistance to changes in air temperature and precipitation than those in the Jura and Swiss Plateau, which





behave similarly to those in Czechia. In general, southern and western mountain ranges experienced larger RoS decreases in both regions.



Figure 5: **Percent of RoS days that occur due to the temperature (T) increase and precipitation (P) changes in selected combinations, compared to reference scenario T0_P1. Boxplots represent the variation among catchments located in the individual mountain ranges, with the 25th and 75th percentiles within a box, the median as a thick line and the whiskers representing maximum and minimum values.**

### 3.4 Seasonal RoS changes for different climate projections

Results showed that changes in RoS occurrence will likely differ considerably for different months of the cold season. A notable RoS increase was detected in January and February across the Swiss catchments (Fig. 6d) and this mid-winter trend was more pronounced than in Czechia. In these winter months, only two projections resulted in a slight RoS decrease in Switzerland. Across the Czech catchments, a RoS increase was limited only to projections leading to wet conditions and a moderate increase in air temperature (Fig. 6b).

Towards the end of the winter, with an earlier snowmelt period onset in Czechia, a decrease in RoS days was simulated for most projections (Figs. 6a and 6b). Projections leading to the temperature increase by more than 2 °C led to a slight RoS



increase only for the wetter months of January or February. Only a few projections, representing wetter conditions and a temperature increase limited to 2 °C, predicted an increase in the number of RoS for the Czech catchments for other months

during the cold season (November, December, March, and April). A similar pattern, although for more projections and with a more substantial RoS increase, was simulated across the Swiss region (Figs. 6c and 6d). This RoS increase resulted from the compensating effect of increased precipitation for projections with a moderate increase in air temperature.

**Figure 6: Absolute (left column, a and c) and a fraction of the number of RoS days (right column, b and d) for all projections (T**
**and P combinations) in both Czech (top row, a and b) and Swiss (bottom row, c and d) regions. Note that changes are related to reference conditions T0_P1 and selected months (May-Oct for CZ, Aug-Oct for CH) are not shown here due to having only a few RoS days.**



## 3.5 RoS changes across elevation zones

Elevation-based differences in the occurrence of RoS days (Fig. 3) were further analyzed in more detail. We identified

considerable patterns in RoS variations across elevation zones (Fig. 7), evaluating all RoS days expected to occur in the study catchments (Figs. 7a and 7c). The wettest projection with no temperature change (T0_P12) is the only projection that suggested a RoS increase for all elevations in both geographical regions (Figs. 7b and 7c). All other projections simulated RoS decline below 1000 m a.s.l. for both study regions, whereas the decline for the Swiss catchments is more pronounced below this elevation level. More than an 80 % decrease may occur below 600 m a.s.l. for the Czech catchments for the

projections with the highest air temperature increase (T4). A similar relative decrease in RoS days occurs for the Swiss catchments at elevations even below 1000 m a.s.l. Another difference between the two study regions was indicated at the highest elevations in Czechia (above 1300 m a.s.l.), where the warmest and wettest projections suggested the RoS increase. In contrast, such an increase was not seen in the simulations for the Swiss catchments.

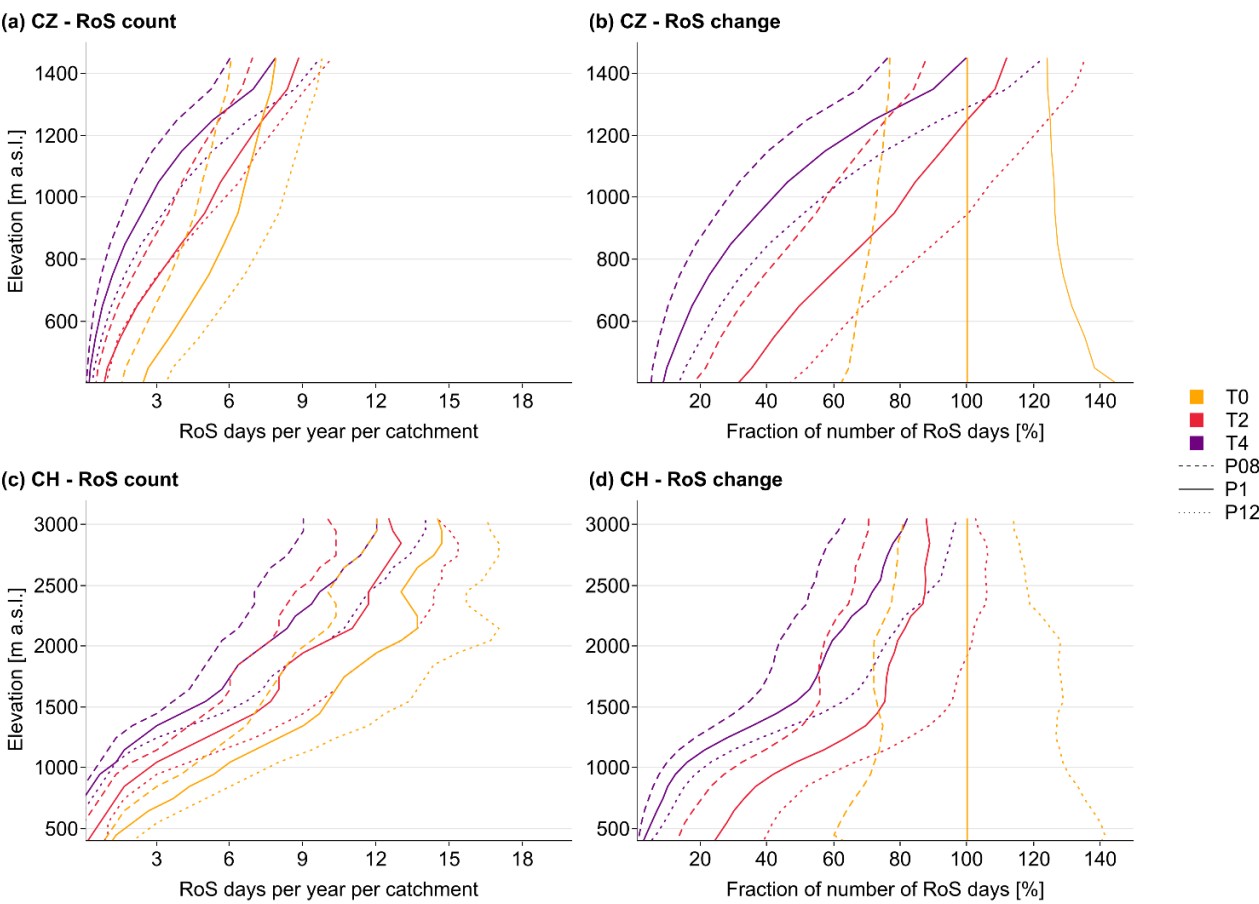

**Figure 7: RoS day occurrence (a and c) and a fraction of the number of RoS days for selected projections compared to reference conditions T0_P1 (b and d) for distinct elevation zones in both Czech (a and b) and Swiss (c and d) regions. The absolute RoS numbers were weighted by the number of catchments within the individual elevation zones. Line colors and styles represent selected temperature (T) and precipitation (P) projections.**



### 3.6 Effect of seasonal characteristics on the RoS occurrence

Several climate and snow parameters defined in Sect. 2.6 were assessed for their relation with the occurrence of RoS days based on the values of Spearman's correlation coefficient (Fig. 8). Results showed some interesting differences between the main regions and individual parameters. Interestingly, a correlation between the occurrence of RoS days and air temperature ($T_{mean}$) was relatively weak (values up to -0.38), but tended to be more negative and thus stronger with projected temperature increase in both regions. Results also indicated that the sum of precipitation ($P_{sum}$) plays a more important role in the

occurrence of RoS days compared to air temperature. In Czechia, relatively strong positive correlations with precipitation totals became less important for warmer projections, which may be associated with the decrease in snowfall totals causing an overall RoS decrease for warmer projections. In Switzerland, seasonal snowfall totals ($S_{sum}$) were detected as the less important driver for RoS occurrence compared to the seasonal sum of all precipitation, regardless of the projection. Parameters related to SWE ($SWE_{mean}$, $SWE_{max}$) were shown as the most important factor for the Czech catchments, showing

an increasing positive relation for the warmer projections.

Similar to the Czech catchments, the importance of SWE increased with the temperature increase in Switzerland. However, the positive correlation was relatively lower, particularly for the projections with a relatively lower increase in air temperature. Snowfall fraction ($S_f$) was identified as the parameter with the largest fluctuations across projections and regions. Positive correlations increasing with the warmer projections were detected in Czechia while increasing snowfall

fraction led to fewer RoS days in projections characterized by a temperature increase of up to 1 °C. Overall, results suggested that RoS events are sensitive to different changes in individual parameters among both regions and individual projections.





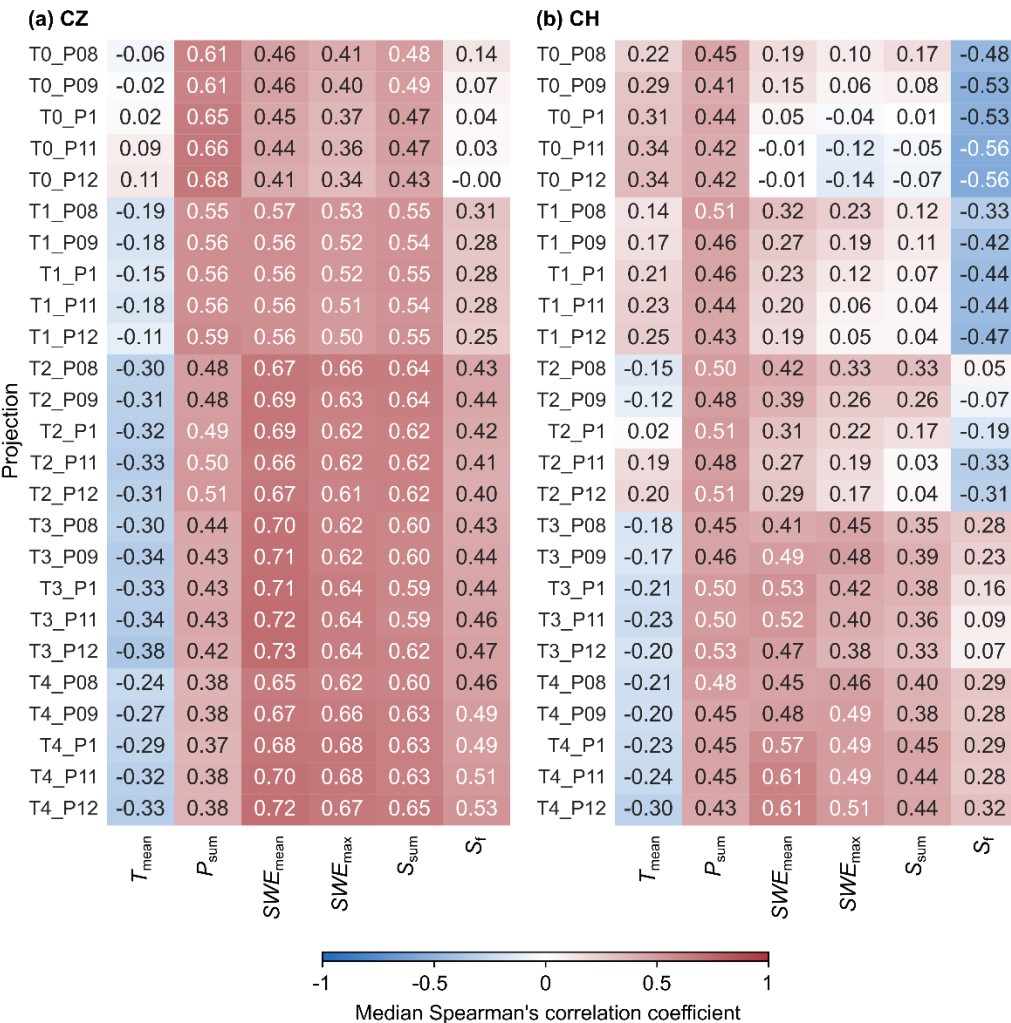

**Figure 8: Median Spearman's correlation coefficients indicated by color and number for all projections in both Czech (a) and**
**Swiss (b) regions valid for the selected climate and snow characteristics: Seasonal mean air temperature ($T_{mean}$), sum of**
**precipitation ($P_{sum}$), mean snow water equivalent ($SWE_{mean}$), maximum snow water equivalent ($SWE_{max}$), sum of snowfall ($S_{sum}$)**
**and snowfall fraction ($S_f$).**

### 3.7 Runoff response to RoS

To assess RoS event-related runoff response and its changes for different projections, the total RoS runoff ($Q_{event}$) for each

RoS event was calculated (Sect. 2.6) and shown as a ratio to the total annual runoff in both regions (Fig. 9). The results show

that RoS-driven runoff contributes to the total runoff with different volumes in both regions, and these contributions are

expected to change for different projections. In the reference conditions (T0_P1), RoS events contributed on average by 10

% to the total annual runoff in the Czech catchments, with the highest contribution of 19 % for some catchments (Fig. 9a). In

Switzerland, where the variability was much higher, RoS events contributed on average 18 % to the total annual runoff with

some catchments contributing up to 35 % (Fig. 9b). The results indicated that the RoS contributions will likely decrease in




the future following a temperature increase. For instance, model simulations suggested that RoS events will be responsible for 5-9 % of the total annual runoff in Czechia for a temperature increase of 2 °C, and 11-16 % in Switzerland. Projections with a temperature increase of 4 °C would reduce the runoff fractions to 2-4 % across the Czech catchments, and 5-9 % for the Swiss catchments. Nevertheless, the RoS runoff decrease caused by the increased air temperature may be partly compensated by the precipitation increase. Despite the expectations that the RoS impact on the total runoff will be lower in the future, extreme hydrological response and flooding triggered by RoS events may still occur.

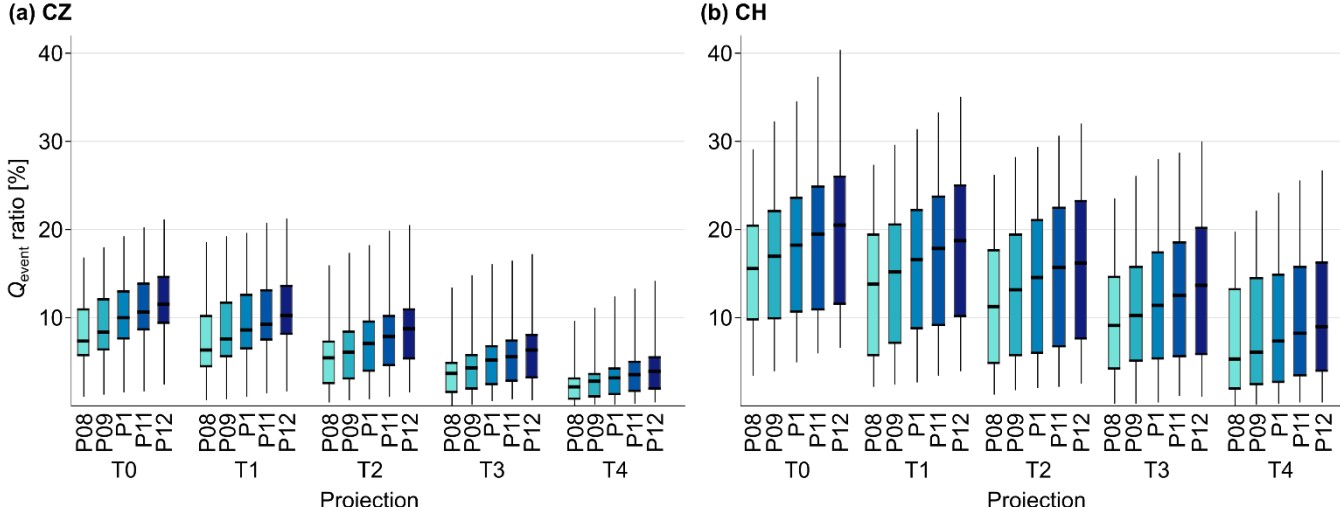

**Figure 9: Fractions of ROS-driven runoff ($Q_{event}$) to the total annual runoff in both Czech (a) and Swiss (b) regions for all projections. Boxes represent the 25th and 75th percentiles, with the median from all catchments indicated as a thick line. Whiskers represent maximum and minimum values. Boxes are grouped and colored according to the temperature (T) and precipitation (P) projections.**

The regional differences in the annual RoS runoff fractions were further investigated for individual months during the cold season, showing the relative changes in the direct runoff during RoS days ($Q_{direct}$) for all projections (Fig. 10). These relative changes, in parallel to the changes in hydrological response, were consistent with relative changes in the number of RoS days shown in Figure 5. For the Czech catchments, a relative increase of at least 25 % was projected only for two projections (T0_P11 and T0_P12) for all months (Fig. 10a). Note that May to October (for Czechia) and August to October (for Switzerland) were excluded from the analysis due to a low number of RoS events, which does not allow for a robust analysis.

In Switzerland, changes in RoS-related direct runoff and RoS occurrence correlated even better, and hydrological impacts generated by RoS events were generally more pronounced (Fig. 10b). In contrast to the Czech catchments, the mid-winter months (December to March) were assessed to be the most hazardous for increased RoS-related runoff response. A notable direct runoff increase of more than 50 % was projected for warmer and wetter projections throughout December to February. For January and February, higher RoS-related runoff was predicted even for some drier projections. With the expected more



frequent RoS events during these months, Swiss catchments, particularly high-elevated ones, may face more extreme RoS-
345 related flood events in the future.

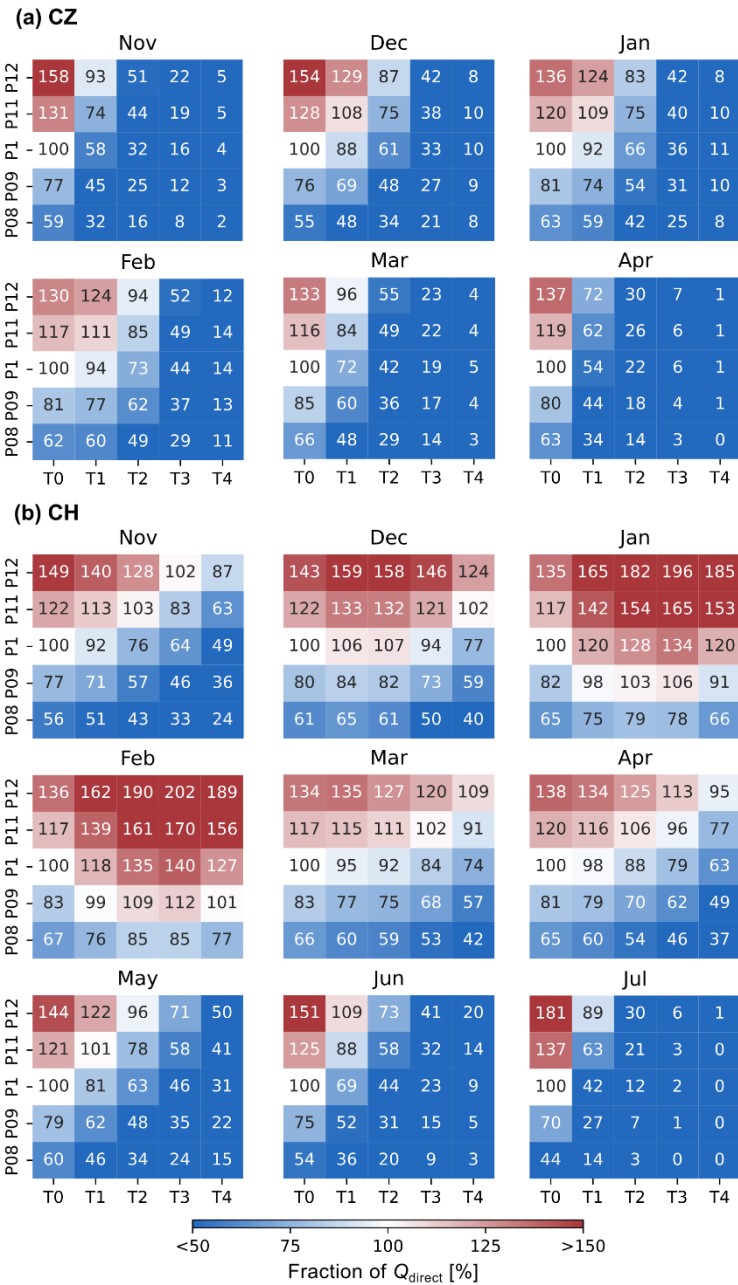

**Figure 10: Monthly fraction in RoS-driven direct runoff (Q_direct) relative to reference conditions T0_P1 (100, white entries, first column, third row) for all projections (T and P combinations) in both Czech (a) and Swiss (b) regions.**



## 4 Discussion

### 350 4.1 HBV model uncertainty

To determine rainfall-runoff components and thus to identify RoS days, RoS events, related variables, and all projected changes, a semi-distributed hydrological model HBV was employed in this study, similar to Freudiger et al. (2014), Juras et al. (2021) and Hotovy et al. (2023). Model calibration, validation and testing were performed in the previous studies using similar datasets (Jenicek and Ledvinka, 2020; Jenicek et al., 2021). Consistently with these studies, multi-criteria model 355 calibration and reiterated calibration runs were performed to reduce the overall parameter uncertainty. Nash-Sutcliffe efficiency values over 0.7, which were also reached for the extended dataset in this study, represent one of the acceptable test criteria (Moriasi et al., 2015).

Since the presented results are based on modeled SWE to define RoS situations, uncertainties arising from the model parametrization need to be addressed. The assessment of the model's ability to simulate SWE and thus detect RoS days 360 correctly was investigated by Hotovy et al. (2023), who compared counts of observed and simulated RoS days, as well as simulated runoff and SWE during RoS events specifically, and did not find major inconsistencies in the model runs and assumed that the model provided sufficiently good simulations. Differences between observed and modeled values may result from the lack of SWE measurements and representativeness of the measurement location, particularly across the Czech catchments. More detailed testing of SWE simulations for the Czech catchments was carried out by Jenicek et al. 365 (2021) and Nedelcev and Jenicek (2021). For example, Nedelcev and Jenicek (2021) compared simulated and observed trends in air temperature, precipitation, and SWE, concluding that the model can provide overall reliable simulations of the above variables, which are temporally and spatially consistent with observed data. SWE simulations were not explicitly evaluated for the Swiss catchments. However, overall model performance is better in simulating SWE for the Swiss catchments, since daily gridded SWE data combining snow depth stational data and a snow density model has been used for 370 model calibration (Magnusson et al., 2014).

The HBV model uses the modified degree-day approach, which may raise further questions about model simplification. However, this simplified method, which is based on a near-linear relationship between snowmelt and air temperature, was hard to outperform at a catchment scale using more sophisticated models, accounting for the entire energy balance of the snowpack (Seibert and Bergström, 2022). Moreover, the complex energy balance approach demands specific data, which is 375 difficult or even impossible to use at a regional scale with various types of catchments. Despite the possible limitations of these bucket-type approaches, several studies have demonstrated that the degree-day method is adequate to be used for snow storage simulation at a catchment scale under a changing climate (Addor et al., 2014; Etter et al., 2017; Juras et al., 2021). Girons Lopez et al. (2020) confirmed that the current HBV snow routine provides results at a catchment level that can hardly be improved despite increasing the physical representation. The above studies confirmed that the model can correctly 380 simulate and distribute all selected snow-related parameters for study catchments adequately to trends in time series. Results



presented in the previous studies showed that model setup, procedures, and derived parameters can satisfactorily represent the actual natural processes, including specifics of RoS events.

## 4.2 RoS definitions

Variations of the threshold values set to identify individual RoS days/events may significantly affect the total number of
recognized situations. A unified RoS definition does not exist in the literature. Different authors use different parameters and thresholds in their studies. The average temperature, duration of snow cover, and the dominant phase of precipitation are expected to be the main factors that explain the variation in the sensitivity of RoS to climate warming (López-Moreno et al., 2021).

As for air temperature, several studies (Bieniek et al., 2018; Crawford et al., 2020; Surfleet and Tullos, 2013) used the
threshold of 0 °C for the daily mean air temperature, while numerous recent studies did not specify the temperature threshold for detecting RoS (Mooney and Li, 2021; Pall et al., 2019; Schirmer et al., 2022; Yang et al., 2022). In this study, we determined the air temperature threshold as one of the RoS-defining parameters, which has been calibrated separately for each of the study catchments. This approach appeared to be a valuable addition to the previous definition used by Hotovy et al. (2023), who used zero as the temperature threshold. The varying threshold temperature may buffer local climatic
conditions affected by different catchment properties, such as elevation range, topography or vegetation, and thus reduce one of the sources of potential errors when identifying RoS days and events. However, we found only minor local differences in the number of RoS days in this presented research as well as in the study performed by Hotovy et al. (2023).

Derived threshold temperatures applied in this study varied from -1.9 to 1.6 °C within all study catchments (Sect. 2.3). This threshold temperature range is comparable to the one presented by Jennings et al. (2018), who identified the temperature
range between -0.4 and 2.4 °C to be valid for 95 % of the stations across the Northern Hemisphere, and indicating the air temperature at which rainfall and snowfall are in equal frequency. Lower temperature thresholds occurred particularly at high-elevated catchments, where snowfall occurs more often than rainfall. The temperature threshold is a challenging criterion that is used in the model to distinguish the phase of precipitation. This can be especially challenging during days when the air temperature fluctuates around the freezing point and consequently, the snowfall fraction is even more sensitive
to the changes in air temperature.

## 4.3 Sensitivity analysis approach

In this study, we investigated potential RoS changes due to variations in climate variables, namely air temperature and precipitation, assessing regional and seasonal changes, future changes at different elevations, and changes in RoS-related runoff response.
To limit uncertainties related to the climatological modeling, a sensitivity analysis was applied in this study instead of the complex climatological modeling approach to assess how air temperature and precipitation changes affect RoS occurrence and extremity. Different sources of uncertainty resulting from the modeling approach were considered in several RoS



studies, with natural climate variability being seen as the primary source of uncertainty in RoS projections (Schirmer et al., 2022). A sensitivity analysis approach for RoS-related research was performed by López-Moreno et al. (2021), who used this method to demonstrate the effects of the warming climate.

In this study, climate variables were altered with regard to the expected future climate variations presented by various respected sources (Gutiérrez et al., 2021). A total of 24 combinations covered a large range of possible future climate behavior. These projections were related to the given reference period, representing the current climate conditions that are already even more than 1 °C warmer compared to the air temperature in the pre-industrial era. Selected temperature projections cover more optimistic ranges with an air temperature increase up to 2 °C but also pessimistic projections with a temperature increase close to 4 °C. Since the direction of precipitation changes is uncertain for the regions of central Europe, projecting both increases and decreases using different climate models, a range from a 20 % decrease to a 20 % increase was applied in this study to cover a wide range in potential future climate.

## 4.4 Observed and future changes in RoS

Studies investigating RoS occurrence are usually limited to specific regions (Li et al., 2019; Yang et al., 2022) since the spatial and temporal distribution of RoS days and events is controlled by current and local weather conditions. Thus, a comparison of RoS occurrence across different regions may be challenging. Notable regional differences within both study regions and individual subregions and mountain ranges were also detected in this study. Local climate variability and uncertainty in climate model projections (discussed more in detail in Sect. 4.3) are other factors that make climate change-driven variations in RoS even more challenging to assess. The sensitivity of RoS to climate change is highly variable among sites and also with different elevations, aspects, and slopes in each basin (López-Moreno et al., 2021).

Our results were consistent with the conclusions presented by Schirmer et al. (2022) or Mooney and Li (2021), who found climate change signals towards more intense and frequent RoS events for an RCP 8.5 scenario (closest to the warmest projections in our study assuming an increase in temperature of 4 °C) at high elevations. Many studies (Il Jeong and Sushama, 2017; Li et al., 2019; Mooney and Li, 2021; Musselman et al., 2018; Sezen et al., 2020; Trubilowicz and Moore, 2017) evaluating and modeling RoS events for different climate scenarios predict an increase of RoS events, particularly at higher elevations (usually valid for catchments above 1500 m a.s.l.). In contrast, their results showed a general RoS decrease with lower hydrological extreme responses at lower elevations (usually covering catchments below 1000 m a.s.l.). These broader elevation-based behaviors were more pronounced in the Czech catchments in our study. Results also showed seasonally-dependent changes in RoS occurrence. Most of the projections suggested a decrease in the number of RoS days towards the end of winter (particularly April and May), which supports the findings presented by Sezen et al. (2020). The signals towards more frequent RoS events, more pronounced in the Swiss catchments, were detected in the middle of the snow season. This RoS increase is likely driven by changes in precipitation since more precipitation is expected to occur as rain rather than snow (Nedelcev and Jenicek, 2021).



There is still limited knowledge on how RoS-driven hydrological response will be affected by climate change (Myers et al., 2023). Therefore, RoS-related runoff projections presented in this study are very beneficial. Sikorska-Senoner and Seibert (2020) identified an overall decreasing trend of RoS-related flooding for 27 Swiss catchments between 1980 and 2014, which agrees with our general results for the Swiss study catchments and throughout the entire year. In our study, we found that these general trends may not be present for winter months (January, February and March) due to expected changes in air

temperature and precipitation patterns. Beniston and Stoffel (2016) concluded that the frequency of floods triggered by RoS may increase by 50 % in Switzerland with a temperature increase of 2-4 °C. However, an air temperature increase of more than 4 °C may result in a RoS-driven flood decrease due to the decline in snowpack duration.

    Runoff projections presented in this study did not specifically assess changes in the extreme hydrological RoS-related response. Thus, these possible climate-driven changes remain uncertain. Such extreme hydrological events triggered by RoS

may occur. However, the probability will likely be lowered with gradual warming, although significant RoS runoff events remain an important flood risk, especially for moderate warming up to 2 °C compared to the reference period. The relative increase in RoS-driven runoff was projected to be even less frequent than the relative increase in the number of RoS days in January and February. This fact may indicate that more frequent RoS occurrence does not necessarily result in increased runoff with potential flooding. All projections with a temperature increase above 2 °C, which seem realistic for the future

climate, show an expected decrease in RoS runoff response for all months.

    According to López-Moreno et al. (2021), the hydrological importance of RoS is not expected to decrease, although the overall frequency of RoS drops. Their model runs showed that maximum runoffs caused by RoS may increase due to warmer snowpack during future RoS events, and more accelerated snowmelt enhanced by energy inputs. The above-mentioned inconsistency between relative changes in RoS numbers (Fig. 6) and relative changes in runoff response (Fig. 10) was

evident in our analyses. The initial snowpack properties and related snowpack retention capacity can also play an important role in runoff formation during RoS events (Garvelmann et al., 2015; Würzer et al., 2016). Consequently, some RoS events do not increase runoff (Juras et al., 2021; Wayand et al., 2015).

## 5 Conclusions

    We evaluated potential regional and seasonal variations in RoS occurrence that are projected to occur in the future due to

climate change. We performed a sensitivity analysis using a conceptual hydrological model simulating the change in RoS situations and their runoff response to precipitation and air temperature changes. Based on the results, we can draw the following conclusions.

    The mean number of RoS days per season varied from one to more than 50 RoS days at a catchment scale, with the most frequent RoS occurrence in the elevation range from 1000 to 2000 m a.s.l. Regarding the elevation, distinct catchments saw

the average RoS occurrence at different times of the year from mid-January to mid-May. March was the month with the highest RoS occurrence.

The results showed that climate change-driven RoS changes are highly variable over regions and sub-regions, across elevations, and within the cold season. In general, RoS days are expected to occur less frequently with further warming, particularly at lower elevations. The warmest projections suggested a decrease of RoS days by about 75 % for the Czech catchments. High-elevation Swiss catchments may respond less sensitively, at least in projections leading to wetter conditions, compared to the reference period. However, the number of RoS days may increase, specifically during the mid-winter (January, February) and at higher elevations following moderate warming, which may be further enhanced by increased precipitation.

Various seasonal climate and snow characteristics may control RoS occurrences. The RoS occurrence was identified as more sensitive to changes in snowfall in the Czech catchments, while seasonal precipitation totals (regardless of snowfall or rainfall) appeared to be the primary driver in Switzerland. Surprisingly, the correlation between RoS and air temperature was relatively weak in both regions.

The results suggested that RoS contribution to annual runoff will likely be reduced from the current 10 % to 2-4 % for the warmest projections in Czechia, and from 18 % to 5-9 % in Switzerland. However, the RoS contribution to runoff may increase in winter months in Switzerland, for almost all projections with the same or higher amount of precipitation, regardless of air temperature increase. With the expected more frequent RoS events during these months, Swiss catchments, particularly high-elevation ones, may face more extreme RoS-related flood events in the future. For Czech catchments, the winter runoff increase is expected only for wet projections with a relatively small air temperature increase. Despite the expectations that the overall RoS impact on runoff will be lower in the future, extreme hydrological response and flooding triggered by RoS events may still represent a significant flood risk.

**Data availability**

The HBV model outputs were published as an open dataset (Jenicek et al., 2024).

**Author contribution**

OH and MJ initiated the study and designed the research; OH, ON and MJ prepared and analyzed data; OH and ON produced the figures and tables; all authors contributed to interpreting the results; OH wrote the manuscript with the contribution of all co-authors.

**Competing interests**

At least one of the (co-)authors is a member of the editorial board of Hydrology and Earth System Sciences.



**Disclaimer**

Publisher's note: Copernicus Publications remains neutral with regard to jurisdictional claims made in the text, published maps, institutional affiliations, or any other geographical representation in this paper. While Copernicus Publications makes every effort to include appropriate place names, the final responsibility lies with the authors.

**Acknowledgments**

The authors wish to thank Tobias Jonas for preparing and providing SWE data for the Swiss catchments. Many thanks are
due to Tracy Ewen for improving the language.

**Financial support**

This work was supported jointly by the Czech Science Foundation and Swiss National Science Foundation (project MountSnow, no. GAČR 23-06859K and SNSF 200021E_213165), Charles University Grant Agency (project no. GAUK 316821), Technology Agency of the Czech Republic (project SS02030040) and the Johannes Amos Comenius Programme
(P JAC; project No. CZ.02.01.01/00/22_008/0004605; Natural and anthropogenic georisks).

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
