# Peer review of "Rain-on-snow events in mountainous catchments under climate change"

_EGUsphere, 2024_

## Author Comment (AC1)

**Author's response to the Reviewer 2**

Black text: Referee comment

Blue text: Authors' response

This manuscript examined rain-on-snow (ROS) frequencies, runoff responses, and relationships to hydroclimate variables across Czechia, Switzerland, and Germany. This was accomplished using a hydrologic model and perturbed runs were also examined to simulate climate change impacts to ROS.

This was a well written paper and fits well into this journal. The results are summarized nicely, and the discussion does well highlighting caveats and uncertainty. My only major comment is around more clarification needed for some of the methods. Otherwise, mostly minor comments. I'm giving it a minor revision as I don't think the "major comment" will take all that much work.

Thank you for the review of our manuscript. We greatly appreciate your constructive comments and suggestions. Please find our point-by-point response below (in blue).

**Major comments**

Several aspects of the data and methods around the HBV model were unclear and could be improved by adding more details. One concern I have that I don't think was addressed was using station-based data as model inputs for the Czech catchments but then using gridded data for the Swiss catchments. Can the authors comment on what impact the station-based vs. gridded inputs might have on the simulations? Also, what was the spatial resolution of the gridded data used? It also was not clear to me what the HBV model inputs were. The historical dataset consists of temperature, precipitation, runoff, and SWE. Were all of these input into the HBV model?

Thank you for this relevant comment. Indeed, different types of input model data (gridded vs. stational) have been used for Czech and Swiss catchments. According to the results presented in Fig. 2, the model performs slightly better for Swiss catchments, where the gridded data has been used. This implies the model should perform better for Czech catchments if gridded data is used. However, only stational data were available at the time of study elaboration. Nowadays, official gridded products are available in Czechia as well, which would potentially enable the model calibrations to be redone using these input data. Although we are planning to do so, this is instead a long-term run and, unfortunately, can't be done for this study.

Nevertheless, we previously used model simulations using stational data in some of our recent studies (e.g., Jenicek and Ledvinka, 2020; Nedelcev and Jenicek, 2021; Hotovy et al., 2023) and did not find any major inconsistencies or errors in the resulting simulations. In the above studies, we also did several tests of the model performance, including its ability to simulate SWE (Nedelcev and Jenicek, 2021) or rain-on-snow events (Hotovy et al., 2023) correctly. These above tests also showed that although the absolute values of runoff signatures may change when different forcing data are used, we expect that the relative differences between individual projections/perturbations (as the major approach used in our study) remain similar for different calibration runs. Thus they will not affect the interpretation of the overall results.

In the revised manuscript, we will add more explanations and discuss this issue. Additionally, we will provide information about the spatial resolution of gridded data (2km for air temperature and precipitation, and 1km for SWE) and better describe the HBV model inputs.

**Other comments**

Figure1: I think this could be improved. As is, the polygons are very small and hard to see. I might suggest and panel plot with several zoomed maps that also include more detailed terrain.

We agree that it is not easy to find the best way to visualize such data. We will prepare some alternatives to the figure and decide which fits best. Besides the reviewer's suggestion, we could make two different maps separately for Czechia and Switzerland combined in one figure.

I would suggest using the term "perturbations" instead of "projections" throughout the paper. Projections are usually associated with GCM outputs in the future, and the future periods are not directly accounted for in your method.

Thank you for your suggestion. Although we have discussed earlier what terminology should be used here, we will consider your suggestion as we are aware that the term "projections" is consistently used rather for climate model outputs.

Line 190: What is the median objective function? Is a value of 1 a perfect score? Please explain.

Yes, a value of 1 represents a perfect fit. We will consider reformulation in the revised manuscript to be more precise.

Figure 3: Great figure! Is the annual number of ROS days the mean? Median? Please specify in the caption.

Thank you. The annual number of RoS represents the annual mean from 1980-2010. We will specify the figure caption in this respect to avoid confusion.

Figure 7: It looks like the T0_P12 line is drawn wrong in 7b. It shows a solid line and based on the text in lines 276-277 it should be small-dashed.

Thank you for double-checking the figure. Indeed, this is the mistake which we will correct in the revised manuscript.

**References**

Hotovy, O., Nedelcev, O., & Jenicek, M. (2023). Changes in rain-on-snow events in mountain catchments in the rain–snow transition zone. *Hydrological Sciences Journal*, *68*(4), 572–584. https://doi.org/10.1080/02626667.2023.2177544

Jenicek, M., & Ledvinka, O. (2020). Importance of snowmelt contribution to seasonal runoff and summer low flows in Czechia. *Hydrology and Earth System Sciences*, *24*(7), 3475–3491. https://doi.org/10.5194/hess-24-3475-2020

Nedelcev, O., & Jenicek, M. (2021). Trends in seasonal snowpack and their relation to climate variables in mountain catchments in Czechia. *Hydrological Sciences Journal*, *66*(16), 2340–2356. https://doi.org/10.1080/02626667.2021.1990298

---

## Author Comment (AC2)

**Author's response to the Reviewer 1**

Black text: Referee comment

Blue text: Authors' response

In this manuscript, the authors evaluate the sensitivity of rain-on-snow (RoS) events and their associated contribution to annual runoff given incremental changes to annual temperature and precipitation magnitudes. They test this by perturbing meteorological conditions from a 30-year reference period (1980-2010) over 93 catchments focused in Czechia, Germany, and Switzerland. This study finds that changes to RoS events depended on the geographic region, with approximately 75% of the lower-elevation Czech catchments demonstrating decreases in RoS with the larges (4 C) increases in temperature. The higher-elevation Swiss catchments were less-sensitive overall, with a higher frequency of unique temperature and precipitation perturbations driving increases to the number of RoS events. Interestingly, the results also show slightly different physical relationships between RoS days and multiple climate and snow variables in the two regions, suggesting different physical drivers of RoS occurrence. Finally, the authors go further by demonstrating how climate-driven changes to RoS events will alter historical RoS contributions runoff, both seasonally and annually.

I would like to commend the authors on their hard work and interesting manuscript. It was clear to me that a considerable amount of work went into this study, and I believe that the methodology, results, and presentation would lend itself well to the scope of HESS. However, I had two main concerns about the study. First, while the study demonstrated interesting sensitivities to incremental changes to both temperature and precipitation, there were a number of methodological and modeling decisions that could be influencing the main results. These include: a precipitation thresholding approach that excludes mixed-phase precipitation, the snow-state requirements used to prescribe the occurrence of a RoS event, the likelihood of unique combinations of temperature and precipitation perturbations, and the assumption of stationary changes in climate. While I don't think it's necessary for an investigation and discussion of these modeling decisions to be a major part of the manuscript, more should be done to establish whether the impacts on RoS and runoff presented by this research are outside the uncertainties that could be driven from the model decisions listed above. Secondly, not enough information on the change to snow cover was included, making it difficult to determine the extent to which RoS frequency was altered by changes to snow cover duration, relative to changes in precipitation phase. A deeper discussion on both of these comments are included in the "Major comments" below.

My recommendation is that this manuscript be returned to the authors for major revisions. Again, I would like to thank the authors for their contribution, and I really enjoyed reading this study. I would be delighted to review this manuscript again if the authors choose to continue with HESS.

Thank you for the review of our manuscript. We greatly appreciate your constructive comments and suggestions. Please find our point-by-point response below (in blue).

**Major comments**

Introduction: The authors do a really nice job at gathering and citing a number of the most relevant studies on RoS. However, the introduction often alludes to findings, complexities, shortfalls, and

uncertainties in these studies without providing details or examples. The authors should consider using some of the literature review to clearly indicate the specific gaps to be addressed by this study. As an example, some of these things that could have been addressed more explicitly include: the list of "unsolved problems" in line 35, the processes/interactions that make RoS events and the resulting hydrology "complex in nature" (line 47), a deeper discussion of what is meant by "compound effect" (compounded uncertainties from estimates of snow cover and predictions of rainfall?, line 48), and what explicitly is included in the "different climate variables" (line 65).

We agree with your comment and will revise the introduction and expand this section to provide more details of the referenced studies and clarify the gaps addressed by our study. The particular changes will include better highlighting the novelty of our study, explaining the main drivers of the RoS variability (the complexity of the process) and the role of climate variables (air temperature and precipitation), which, together with snowpack, jointly influence changes in the frequency of ROS events and their runoff responses.

As noted by the authors, RoS occurrence and severity depends on both snow cover existing, and warm temperatures coinciding with a precipitation event. Given this, more should be done throughout the manuscript to establish that the results are statistically-significant provided the uncertainties from the experimental setup. For example:

1.  My understanding of the modeling setup is that although the threshold used to partition rain and snow could change based on the basin, it didn't allow mixed-phase precipitation meaning that all of a given timestep's precipitation fell as snow or rain given the smallest of changes in temperature across any given temperature threshold. Since mixed-phase precipitation can often occur across large spreads in temperature, do the authors know how the decision to use a static temperature threshold impacted the frequency of RoS relative to a threshold that allowed mixed-phase rain and snow? My hypothesis is that allowing mixed-phase while keeping the decision to filter drizzle (daily precip < 5 mm) may result in less-sensitive changes to RoS in response to changes in temperature.

Thank you for raising this important point. You are correct that our approach does not allow mixed-phase precipitation, as we applied a temperature threshold (TT) to classify precipitation as either rain or snow. The parameter TT was calibrated in the model separately for each catchment. To address your comment, we will perform supplementary analyses to test how different TT values (e.g., within some range around the calibrated value) influence the number of RoS days. We hypothesise that the absolute number of RoS days will change, but the relative variability and trends will remain unchanged, including relative changes in RoS days for different perturbations. Besides this additional analysis, we will expand the discussion section by adding some studies that tested the influence of different TT values on snow routine simulations including different snow/rain separation approaches. For example, a study by Girons Lopez et al. (2020) tested several modifications of the HBV model snow routine in Swiss and Czech catchments (a subset of those used in this study), including those assuming different snow/rain separation approaches. The study showed that the snow routine currently employed in the HBV model provided relatively good results, and none of the tested modifications resulted in substantially increased model performance.

2.  I like the authors' decision to filter regions where RoS occurred based on a SWE state threshold of 10 mm. However, it's worth noting that larger footprints of shallow snow experiencing a RoS event may still contribute significantly to the hydrologic pulse. This may be particularly true in future climates, and across shallow-snow regions which melt more-readily and rapidly with the heightened turbulent, latent, and sensible heat fluxes during RoS events. Did the authors test other SWE thresholds?

We agree that shallow snow can contribute significantly to hydrological responses during RoS events, particularly in future climates where shallow snow is expected to occur more often and will be more susceptible to rapid melting. In our analysis, we chose the 10 mm SWE threshold to ensure that RoS days/events were defined in areas with meaningful snow cover. However, this 10 mm SWE threshold represents a catchment mean. Thus, it covers a variety of possible snow distribution, such as equally distributed snow cover across the catchment or higher snowpack at high elevations and no snowpack at lower elevations. As we did not explicitly test other SWE thresholds in this study, we will now make a simple supplementary test of different SWE thresholds (5 mm, 20 mm) to see how the threshold is sensitive to the absolute value of identified RoS days (similarly as described in point 1 above). Additionally, we will better discuss this issue in the discussion section.

3. The combinations of the precipitation and temperature perturbations are presented as if each of these are equal-likelihood. While I like this structured investigation of impacts from incremental changes to both temperature and precipitation, there are some edge-case scenarios that may be less likely given projected changes to climate (e.g., T4_P08). The authors should consider adding some text to ground which combinations of temperature and precipitation changes, and resulting changes to RoS frequency and severity, may be more and less-likely to emerge in future climates.

We agree that not all combinations of tested perturbations are equally likely to occur under projected climate. We will discuss this issue carefully and add some text to indicate which perturbations are closer to the expected future climate (e.g., +2 and +3°C and P09, P1 and P11). However, we should also emphasize more that the methodological approach we used in our study was designed to systematically investigate the sensitivity of catchment response to changes in climate variables (air temperature and precipitation) rather than project changes in RoS in future climate.

4. My overarching largest concern is the assumption of stationary changes in climate. For example, I believe this modeling setup applied a constant multiplier to the historical precipitation record, preserving the timing, severity, and frequency of precipitation, and how it aligned with swings in temperature. Provided the fact that both winter precipitation magnitude and frequency is expected to change in future climates (and more so in the winter than the summer), and that these may be more likely to coincide with moist and warm temperatures, it's likely that an increase to the frequency of precipitation events may overwhelm some of the changes to RoS frequency and severity driven by stationary changes to temperature and precipitation. The authors should investigate this.

We are aware that our assumption of climate stationarity simplifies the complexities of future climate changes, particularly the shifts in precipitation patterns and their interactions with temperature. However, as also mentioned in our previous response, our analyses should not be interpreted as the projection of future climate. We will improve the discussion regarding these limitations and our modelling setup.

While some of the points from above are mentioned briefly in the study discussion, the authors should consider expanding on them to investigate where, when, and in what cases the RoS sensitivities reported by this study fall outside of the noise expected from the experimental setup and procedure. The authors could consider test cases using the full range of years and catchments, or case-studies based on catchments comparing the least (e.g., coldest and driest) and most-sensitive (e.g., catchments within the transition zone) locations.

Thank you for your suggestion. We consider presenting some results just for a few selected catchments with contrasting characteristics and for contrasting years (cold/warm, wet/dry). We will

present the results in a supplement. This will enhance the robustness and relevance of our study and address your concern effectively.

Except for Figure 8, there is little presented about the impact that changes to temperature and precipitation have on simulated snow cover. This is particularly important since changes in RoS frequency can be driven to a first-order by changes to snow cover. Additionally, the model used in this study simulated 100 m elevation bands, thereby assuming full or absent snow cover for the full band while some level of fractional snow cover likely existed. Many of the projected decreases to RoS frequency are consistent with the signals expected for a reducing snow cover. However, this isn't presented explicitly. There are also some results that suggest that there may be RoS increases driven by an increase in the snow cover duration. For example, Figure 5 shows an increase (relative to the historical) in the percent of RoS days from T2_P1 to T2_P12 in the Western Sudetes. Given 1) that only the magnitude and not the frequency/timing of precipitation events aren't changing between T2_P1 and T2_P12, and 2) temperature is not changing between T2_P1 and T2_P12, the ~25% increase in RoS frequency between these two models must have been driven by changes in snow cover. Is that correct? Are these increases in RoS frequency happening earlier or later in the snow season, and across what elevation bands?

Thank you for this comment. Snow cover indeed plays a critical role in driving RoS dynamics. Therefore, it is worth adding more about linking changes in snow cover to RoS frequency to describe these connections better. To show this, we will take all precipitation days and distinguish them into four categories: 1) precipitation is classified as snowfall falling on snow-free ground, 2) precipitation is classified as snowfall falling on existing snow cover (>10 mm SWE), 3) precipitation is classified as rainfall falling on the snow-free ground, and 4) precipitation is classified as rainfall falling on existing snow cover. In the next step, we will examine how these categories change for individual perturbations. This approach enables us to calculate, e.g., whether the changes in RoS days (category rain on snow cover) are driven by disappearing/advancing snow cover or by changes in the precipitation phase.

Section 3.7: I really like this analysis on the runoff response! This is a great advancement on the field of RoS studies.

Thank you for the positive comment. The runoff response analysis was indeed one of the main aims of our study and we believe it brought some novel results to our study.

**Minor comments**

Line 8: Change "increase" to "increases" in order to match the tense of "changes".

It will be changed, thank you.

Line 8: "Occurrence" is used throughout to reference RoS events. The authors could consider revising "snow occurrence" to "snow cover" in this context (also line 51).

We agree, and we will change the wording.

Line 10: Delete "were evaluated"

It will be deleted.

Lines 34 – 36: Without more context provided for the "unsolved problems", this paragraph doesn't provide a lot of new information, especially considering the literature review provided in the

following paragraphs. I think the authors could consider removing these lines. I would have the same comment for the last line of the following paragraph: "Much of the current research … under ongoing climate change".

We will consider removing the text or providing more context.

Line 57: What is meant by "spatial and temporal distribution"? Is this referencing the changes to snow cover in time and space, in addition to the changes to rainfall frequencies?

Spatial and temporal distribution of RoS occurrence was meant here. We will change the wording to be more precise.

Lines 100 – 101: In instances where a station came from outside of the catchment bounds, how far away were these stations, on average? What was the maximum and minimum distances, and what sort of uncertainties could be expected for the stations that are the furthest away?

You are correct that, in some cases, the weather stations used to drive the HBV model were located outside the catchments. However, this issue is relevant only for some Czech catchments since grided data were used for Swiss catchments. If such a situation occurred, we tried to select the closest station possible (usually up to 10-20 km from the catchment border) located within the elevation range of the specific catchment. The distances were usually shorter for precipitation than for air temperature since the network of gauging stations measuring only precipitation is denser than that of fully equipped climate stations. We will expand this section in the revised version with additional information above.

Line 109: Is there a citation for the MeteoSwiss gridded meteorology product? What is the spatial resolution of this forcing?

The precipitation and temperature data is in 2km spatial resolution, and the SWE data is in 1km resolution. We will add this information, together with the correct reference, to the revised version.

Line 144: Replace "e.g. Hotovy et al. (2023)" with "(Hotovy et al., 2023)".

We will replace it.

Lines 154 – 156: I'm finding this passage a little confusing. The authors specify that RoS events are "multi-day" events, but then follow by saying that RoS events can include both RoS days and non-RoS days. Does this just mean that the runoff can peak after the date of the actual rainfall?

Yes, you are right. The idea was to define these two situations separately, to distinguish between causing events (days with rain) and subsequent runoff responses, which usually last longer. We will rewrite the text to be clearer for readers.

Line 166: Why does P08 correspond with -20%? Can these just be named based on their percent-perturbations (e.g., P-20% or P80%)?

Thank you for your suggestion. We considered different ways of naming the individual perturbations while writing the original manuscript, realizing that there are certainly more equivalents. The P08 perturbation (for example) represents 0.8 multiplication of the original precipitation in the reference period (P0T1). Nevertheless, we will consider the naming options again.

Figure 3: I'm having a difficult time distinguishing the different colors, and especially in instances when the points are separated further in space. The authors could consider a few things: 1) testing different color bars, 2) adapting both the point color and size to correspond with the average RoS day occurrence, and/or 3) breaking this plot into two separate subplots -- one with a y-axis corresponding to RoS days per year and the other with the average RoS day occurrence.

Figure 3: How large is the spread in the timing of the RoS day occurrence? If the authors choose to break this into a different subplot, they could consider including whiskers to show this.

Thank you for both suggestions related to Figure 3. We will prepare alternatives as suggested and consider the best and most informative one.

Line 213: This is a huge number of RoS events! Is there evidence to back up that this is grounded in reality?

Unfortunately, we are not able to provide the evidence based on measurements. The absolute values of the simulated RoS days are indeed affected by selected thresholds used for their calculations (such as snow/rain temperature or minimum SWE) as described in our responses to major comments above (we refer to our explanation there). However, in our previous study, Hotovy et al. (2023) (supplementary Fig. S4), we provided a comparison of the number of RoS days for Czech catchments (similar selection as in this current manuscript) calculated using observed and simulated data (although using a single threshold temperature threshold) showing overall good fit between observed and simulated values. Therefore, the number of RoS days may be realistic. It is worth noting that the majority of RoS events do not produce significant runoff response as shown also by Juras et al. (2021) for Czech catchments.

Figure 7b: It looks like there are two solid yellow (T0_P1) lines. My guess is that the rightmost line actually represents T0_P12.

Thank you for this point. We have already realized this mistake, and we will correct this in the revised manuscript.

**References**

Girons Lopez, M., Vis, M. J. P., Jenicek, M., Griessinger, N., & Seibert, J. (2020). Assessing the degree of detail of temperature-based snow routines for runoff modelling in mountainous areas in central Europe. *Hydrology and Earth System Sciences*, *24*(9), 4441–4461. https://doi.org/10.5194/hess-24-4441-2020

Hotovy, O., Nedelcev, O., & Jenicek, M. (2023). Changes in rain-on-snow events in mountain catchments in the rain–snow transition zone. *Hydrological Sciences Journal*, *68*(4), 572–584. https://doi.org/10.1080/02626667.2023.2177544

Juras, R., Blöcher, J. R., Jenicek, M., Hotovy, O., & Markonis, Y. (2021). What affects the hydrological response of rain-on-snow events in low-altitude mountain ranges in Central Europe? *Journal of Hydrology*, *603*, 127002. https://doi.org/10.1016/j.jhydrol.2021.127002

---

## Author Response (AR1)

**Authors' response**

Dear Editor and Reviewers,

Thank you for your valuable comments and suggestions to improve our contribution. We have prepared a revised version of the manuscript that takes into account all the reviewers' comments (see our point-by-point response below). The following major changes have been made to the original manuscript:

- We have carried out additional analyses showing how changes in snow cover affect the frequency of RoS. The results are shown in the new supplementary Figure S1 and discussed in the revised manuscript.
- We tested how climate perturbations affect RoS occurrence in contrasting snow-rich and snow-poor seasons (Fig. S2 in the supplementary material) and how the number of RoS differs in a few selected catchments with contrasting characteristics (Fig. S3 in the supplementary material).
- We performed additional analyses, as suggested by reviewer 1, related to testing the threshold parameters used to define RoS days in our study. These include testing different thresholds for temperature to distinguish between snowfall and rainfall, and thresholds for SWE. The results are discussed in the manuscript and visually presented in this response.
- We have improved the introduction section, mainly by better addressing the research gaps.
- We have provided several text rewrites as suggested by both reviewers.

Thank you for your consideration of the revised manuscript.

On behalf of all co-authors,

Michal Jeníček

**Review 1**

In this manuscript, the authors evaluate the sensitivity of rain-on-snow (RoS) events and their associated contribution to annual runoff given incremental changes to annual temperature and precipitation magnitudes. They test this by perturbing meteorological conditions from a 30-year reference period (1980-2010) over 93 catchments focused in Czechia, Germany, and Switzerland. This study finds that changes to RoS events depended on the geographic region, with approximately 75% of the lower-elevation Czech catchments demonstrating decreases in RoS with the larges (4 C) increases in temperature. The higher-elevation Swiss catchments were less-sensitive overall, with a higher frequency of unique temperature and precipitation perturbations driving increases to the number of RoS events. Interestingly, the results also show slightly different physical relationships between RoS days and multiple climate and snow variables in the two regions, suggesting different physical drivers of RoS occurrence. Finally, the authors go further by demonstrating how climate-driven changes to RoS events will alter historical RoS contributions runoff, both seasonally and annually.

I would like to commend the authors on their hard work and interesting manuscript. It was clear to me that a considerable amount of work went into this study, and I believe that the methodology, results, and presentation would lend itself well to the scope of HESS. However, I had two main concerns about the study. First, while the study demonstrated interesting sensitivities to incremental changes to both temperature and precipitation, there were a number of methodological and modeling decisions that could be influencing the main results. These include: a precipitation

thresholding approach that excludes mixed-phase precipitation, the snow-state requirements used to prescribe the occurrence of a RoS event, the likelihood of unique combinations of temperature and precipitation perturbations, and the assumption of stationary changes in climate. While I don't think it's necessary for an investigation and discussion of these modeling decisions to be a major part of the manuscript, more should be done to establish whether the impacts on RoS and runoff presented by this research are outside the uncertainties that could be driven from the model decisions listed above. Secondly, not enough information on the change to snow cover was included, making it difficult to determine the extent to which RoS frequency was altered by changes to snow cover duration, relative to changes in precipitation phase. A deeper discussion on both of these comments are included in the "Major comments" below.

My recommendation is that this manuscript be returned to the authors for major revisions. Again, I would like to thank the authors for their contribution, and I really enjoyed reading this study. I would be delighted to review this manuscript again if the authors choose to continue with HESS.

Thank you for the review of our manuscript. We greatly appreciate your constructive comments and suggestions. Please find our point-by-point response below (in blue). The line numbers in our responses are related to the revised version.

**Major comments**

Introduction: The authors do a really nice job at gathering and citing a number of the most relevant studies on RoS. However, the introduction often alludes to findings, complexities, shortfalls, and uncertainties in these studies without providing details or examples. The authors should consider using some of the literature review to clearly indicate the specific gaps to be addressed by this study. As an example, some of these things that could have been addressed more explicitly include: the list of "unsolved problems" in line 35, the processes/interactions that make RoS events and the resulting hydrology "complex in nature" (line 47), a deeper discussion of what is meant by "compound effect" (compounded uncertainties from estimates of snow cover and predictions of rainfall?, line 48), and what explicitly is included in the "different climate variables" (line 65).

We agree with your comment and have revised the introduction and expanded this section to provide more details of the referenced studies and clarify the gaps addressed by our study (L 35-37, 47-50, 68-70 of the revised manuscript). The particular changes include better highlighting the novelty of our study, explaining the main drivers of the RoS variability (the complexity of the process) and the role of climate variables (air temperature and precipitation), which, together with snowpack, jointly influence changes in the frequency of ROS events and their runoff responses.

As noted by the authors, RoS occurrence and severity depends on both snow cover existing, and warm temperatures coinciding with a precipitation event. Given this, more should be done throughout the manuscript to establish that the results are statistically-significant provided the uncertainties from the experimental setup. For example:

1. My understanding of the modeling setup is that although the threshold used to partition rain and snow could change based on the basin, it didn't allow mixed-phase precipitation meaning that all of a given timestep's precipitation fell as snow or rain given the smallest of changes in temperature across any given temperature threshold. Since mixed-phase precipitation can often occur across large spreads in temperature, do the authors know how the decision to use a static temperature threshold impacted the frequency of RoS relative to a threshold that allowed mixed-phase rain and snow? My hypothesis is that allowing mixedphase while keeping the decision to filter drizzle (daily precip < 5 mm) may result in less-sensitive changes to RoS in response to changes in temperature.

Thank you for raising this important point. You are correct that our approach does not allow mixed-phase precipitation, as we applied a temperature threshold ($T_T$) to classify precipitation as either rain or snow. The parameter $T_T$ was calibrated in the model separately for each catchment. To address your comment, we performed supplementary analyses to test how different $T_T$ values (within a defined range around the calibrated value) influence the number of RoS days. The results of this analysis are shown in Fig. 1 of this response and discussed in lines 399-405. The results showed that the absolute number of RoS days changed (left panels of Fig. 1). Still, the relative variability and trends remained unchanged for the reference period and individual perturbations (the points mostly overlap in the right panels).

Besides the above additional analysis, we expanded the discussion section and addressed the issue by providing more information on results sensitivity to changes in model parameters (L 410-418). For example, a study by Girons Lopez et al. (2020) tested several modifications of the HBV model snow routine in Swiss and Czech catchments (a subset of those used in this study), including those assuming different snow/rain separation approaches. The study showed that the snow routine currently employed in the HBV model provided relatively good results, and none of the tested modifications resulted in substantially increased model performance.

[Figure]

Fig. 1: Number of RoS days per year in both Czech (a, b) and Swiss (c, d) regions for all projections (a, c), and a fraction of the number of RoS days relative to reference conditions in both regions for all projections (b, d). Points represent mean values from all catchments within each region (CZ and CH) and are colored according to the given temperature threshold ($T_T$).

2.  I like the authors' decision to filter regions where RoS occurred based on a SWE state threshold of 10 mm. However, it's worth noting that larger footprints of shallow snow experiencing a RoS event may still contribute significantly to the hydrologic pulse. This may be particularly true in future climates, and across shallow-snow regions which melt more-readily and rapidly with the heightened turbulent, latent, and sensible heat fluxes during RoS events. Did the authors test other SWE thresholds?

We agree that shallow snow can contribute significantly to hydrological responses during RoS events, particularly in future climates where shallow snow is expected to occur more often and will be more susceptible to rapid melting. In our analysis, we chose the 10 mm SWE threshold to ensure that RoS days/events were defined in areas with meaningful snow cover. However, this 10 mm SWE threshold represents a catchment mean. Thus, it covers a variety of possible snow distribution, such as equally distributed snow cover across the catchment or higher snowpack at high elevations and no snowpack at lower elevations. As we did not explicitly test other SWE thresholds in this study, we newly made a supplementary test of different SWE thresholds (5 mm, 10 mm, and 20 mm) to see how the threshold is sensitive to the absolute value of identified RoS days (see Fig. 2 of this response). Similar to the above $T_T$ testing, the SWE threshold influences the absolute number of RoS days (left panels in Fig. 2). The relative change (right panels) remained almost the same for the reference period (points mostly overlap), although somewhat changed for individual perturbations. However, the general trends of the changes remain unaffected, with an overall first-order decrease in RoS days fractions with air temperature and a particular increase with increasing precipitation.

We added the above results to the discussion section on lines 448-453 of the revised version.

[Figure]

Fig. 2: Number of RoS days per year in both Czech (a, b) and Swiss (c, d) regions for all projections (a, c), and a fraction of the number of RoS days relative to reference conditions in both regions for all projections (b, d). Points represent mean values from all catchments within each region (CZ and CH) and are colored according to the given SWE threshold.

3. The combinations of the precipitation and temperature perturbations are presented as if each of these are equal-likelihood. While I like this structured investigation of impacts from incremental changes to both temperature and precipitation, there are some edge-case scenarios that may be less likely given projected changes to climate (e.g., T4_P08). The authors should consider adding some text to ground which combinations of temperature and precipitation changes, and resulting changes to RoS frequency and severity, may be more and less-likely to emerge in future climates.

We agree that not all combinations of tested perturbations are equally likely to occur under the projected climate. We added more discussion to address this issue carefully by adding text suggesting which perturbations are closer to the expected future climate (+2 and +3°C and P09, P1 and P11 perturbations, see L 474-477). However, we also emphasized more that the methodological approach we used in our study was designed to systematically investigate the sensitivity of catchment response to changes in climate variables (air temperature and precipitation) rather than project changes in RoS in future climate (Section 2.5 and discussion on lines 456-460).

4. My overarching largest concern is the assumption of stationary changes in climate. For example, I believe this modeling setup applied a constant multiplier to the historical precipitation record, preserving the timing, severity, and frequency of precipitation, and how it aligned with swings in temperature. Provided the fact that both winter precipitation magnitude and frequency is expected to change in future climates (and more so in the winter than the summer), and that these may be more likely to coincide with moist and warm temperatures, it's likely that an increase to the frequency of precipitation events may overwhelm some of the changes to RoS frequency and severity driven by stationary changes to temperature and precipitation. The authors should investigate this.

We are aware that our assumption of climate stationarity simplifies the complexities of future climate changes, particularly the shifts in precipitation patterns and their interactions with temperature. However, as also mentioned in our previous response, our analyses should not be interpreted as the projection of future climate. We addressed this issue in the discussion section by explaining the above limitations and our modelling setup (L 454-466 of the revised version).

While some of the points from above are mentioned briefly in the study discussion, the authors should consider expanding on them to investigate where, when, and in what cases the RoS sensitivities reported by this study fall outside of the noise expected from the experimental setup and procedure. The authors could consider test cases using the full range of years and catchments, or case-studies based on catchments comparing the least (e.g., coldest and driest) and most-sensitive (e.g., catchments within the transition zone) locations.

Thank you for your suggestion. To address your comment and to enhance the robustness and relevance of our study, we performed supplementary analyses to test how climate perturbations affect RoS occurrence in contrasting snow-rich and snow-poor seasons (Fig. S2 in the supplementary material) and how the number of RoS differs in a few selected catchments with contrasting characteristics (Fig. S3 in the supplementary material). The results proved site-specific patterns in the RoS frequency. Also, they revealed that RoS days/events are more temperature-sensitive during snow-poor seasons when a notable decrease of RoS days was identified for the warmest perturbations (T3 and T4). In contrast, the number of RoS days in the snow-rich seasons remains similar or even higher for the warmest perturbations, likely due to more frequent RoS occurrence during a prolonged period with existing snow on the ground, with more RoS days/events expected early and late season. The above results are commented on in the revised manuscript (L 490-497).

Except for Figure 8, there is little presented about the impact that changes to temperature and precipitation have on simulated snow cover. This is particularly important since changes in RoS frequency can be driven to a first-order by changes to snow cover. Additionally, the model used in this study simulated 100 m elevation bands, thereby assuming full or absent snow cover for the full band while some level of fractional snow cover likely existed. Many of the projected decreases to RoS frequency are consistent with the signals expected for a reducing snow cover. However, this isn't presented explicitly. There are also some results that suggest that there may be RoS increases driven by an increase in the snow cover duration. For example, Figure 5 shows an increase (relative to the historical) in the percent of RoS days from T2_P1 to T2_P12 in the Western Sudetes. Given 1) that only the magnitude and not the frequency/timing of precipitation events aren't changing between T2_P1 and T2_P12, and 2) temperature is not changing between T2_P1 and T2_P12, the ~25% increase in RoS frequency between these two models must have been driven by changes in snow cover. Is that correct? Are these increases in RoS frequency happening earlier or later in the snow season, and across what elevation bands?

Thank you for this comment. Snow cover indeed plays a critical role in driving RoS dynamics. Therefore, we performed additional analysis, which better shows how snow-cover changes affect the RoS frequency. Specifically, we took all precipitation days and distinguished them into four categories: 1) precipitation is classified as snowfall falling on the snow-free ground, 2) precipitation is classified as snowfall falling on existing snow cover (>10 mm SWE), 3) precipitation is classified as rainfall falling on the snow-free ground, and 4) precipitation is classified as rainfall falling on existing snow cover. In the next step, we examined how these categories changed for individual perturbations. This approach enabled us to calculate, e.g., whether the changes in RoS days (category rain on snow cover) were driven by disappearing/advancing snow cover or by changes in the precipitation phase and amount. The above results are shown in Fig. S1 and discussed in lines 323-334 of the revised manuscript.

Section 3.7: I really like this analysis on the runoff response! This is a great advancement on the field of RoS studies.

Thank you for the positive comment. The runoff response analysis was indeed one of the main aims of our study and we believe it allowed us to obtain some novel results.

**Minor comments**

Line 8: Change "increase" to "increases" in order to match the tense of "changes".

It has been changed, thank you (L 9).

Line 8: "Occurrence" is used throughout to reference RoS events. The authors could consider revising "snow occurrence" to "snow cover" in this context (also line 51).

We agree, we changed the wording (L 9 and 56).

Line 10: Delete "were evaluated"

It was deleted.

Lines 34 – 36: Without more context provided for the "unsolved problems", this paragraph doesn't provide a lot of new information, especially considering the literature review provided in the following paragraphs. I think the authors could consider removing these lines. I would have the same

comment for the last line of the following paragraph: "Much of the current research … under ongoing climate change".

We provided more context for the part with "unsolved problems" (L 35-37) and removed the last sentence of this paragraph.

Line 57: What is meant by "spatial and temporal distribution"? Is this referencing the changes to snow cover in time and space, in addition to the changes to rainfall frequencies?

Spatial and temporal distribution of RoS occurrence was meant here. We changed the wording to be more precise (L 59).

Lines 100 – 101: In instances where a station came from outside of the catchment bounds, how far away were these stations, on average? What was the maximum and minimum distances, and what sort of uncertainties could be expected for the stations that are the furthest away?

You are correct that, in some cases, the weather stations used to drive the HBV model were located outside the catchments. However, this issue is relevant only for some Czech catchments since grided data were used for Swiss catchments. If such a situation occurred, we selected the closest station (usually up to 10-20 km from the catchment border) located within the elevation range of the specific catchment. The distances were usually shorter for precipitation than for air temperature since the network of gauging stations measuring only precipitation is denser than that of fully equipped climate stations. We expanded the methods section in the revised version with the above information (L 107).

Line 109: Is there a citation for the MeteoSwiss gridded meteorology product? What is the spatial resolution of this forcing?

The precipitation and temperature data is in 2 km spatial resolution, and the SWE data is in 1 km resolution. We added this information, together with the names of the related MeteoSwiss products to the revised version (L 115-119).

Line 144: Replace "e.g. Hotovy et al. (2023)" with "(Hotovy et al., 2023)".

We replaced it (L 154).

Lines 154 – 156: I'm finding this passage a little confusing. The authors specify that RoS events are "multi-day" events, but then follow by saying that RoS events can include both RoS days and non-RoS days. Does this just mean that the runoff can peak after the date of the actual rainfall?

Yes, you are right. The idea was to define these two situations separately distinguishing between causing events (days with rain) and subsequent runoff responses, which usually last longer. We reformulated the text to be clearer for readers (L 164-167).

Line 166: Why does P08 correspond with -20%? Can these just be named based on their percent-perturbations (e.g., P-20% or P80%)?

Thank you for your suggestion. We considered different ways of naming the individual perturbations while writing the original manuscript, realizing that there are certainly more equivalents. The P08 perturbation (for example) represents 0.8 multiplication of the original precipitation in the reference

period (P0T1). After the new consideration, we still prefer to keep this naming convention for its simplicity.

Figure 3: I'm having a difficult time distinguishing the different colors, and especially in instances when the points are separated further in space. The authors could consider a few things: 1) testing different color bars, 2) adapting both the point color and size to correspond with the average RoS day occurrence, and/or 3) breaking this plot into two separate subplots -- one with a y-axis corresponding to RoS days per year and the other with the average RoS day occurrence.

Figure 3: How large is the spread in the timing of the RoS day occurrence? If the authors choose to break this into a different subplot, they could consider including whiskers to show this.

Thank you for both suggestions related to Figure 3. We tested several alternatives as suggested. The same composition with different color bars (from red to grey) was the best and most informative option. We hope this change makes the figure more readable.

Line 213: This is a huge number of RoS events! Is there evidence to back up that this is grounded in reality?

Unfortunately, we are not able to provide the evidence based on measurements. The absolute values of the simulated RoS days are indeed affected by selected thresholds used for their calculations (such as snow/rain temperature or minimum SWE) as described in our responses to major comments above (we refer to our explanation there). However, in our previous study, Hotovy et al. (2023), we provided a comparison of the number of RoS days for Czech catchments (similar selection as in this current manuscript) calculated using observed and simulated data (although using a single threshold temperature threshold) showing overall good fit between observed and simulated values. Therefore, the number of RoS days may be realistic. It is worth noting that the majority of RoS events do not produce significant runoff response as shown also by Juras et al. (2021) for Czech catchments. We extended the discussion section with the above information (L 483-486).

Figure 7b: It looks like there are two solid yellow (T0_P1) lines. My guess is that the rightmost line actually represents T0_P12.

Thank you for this point. We corrected this and slightly modified the figure in the revised manuscript (see new Fig. 7).

**References**

Girons Lopez, M., Vis, M. J. P., Jenicek, M., Griessinger, N., & Seibert, J. (2020). Assessing the degree of detail of temperature-based snow routines for runoff modelling in mountainous areas in central Europe. *Hydrology and Earth System Sciences*, *24*(9), 4441–4461. https://doi.org/10.5194/hess-24-4441-2020

Hotovy, O., Nedelcev, O., & Jenicek, M. (2023). Changes in rain-on-snow events in mountain catchments in the rain–snow transition zone. *Hydrological Sciences Journal*, *68*(4), 572–584. https://doi.org/10.1080/02626667.2023.2177544

Juras, R., Blöcher, J. R., Jenicek, M., Hotovy, O., & Markonis, Y. (2021). What affects the hydrological response of rain-on-snow events in low-altitude mountain ranges in Central Europe? *Journal of Hydrology*, *603*, 127002. https://doi.org/10.1016/j.jhydrol.2021.127002

**Review 2**

This manuscript examined rain-on-snow (ROS) frequencies, runoff responses, and relationships to hydroclimate variables across Czechia, Switzerland, and Germany. This was accomplished using a hydrologic model and perturbed runs were also examined to simulate climate change impacts to ROS.

This was a well written paper and fits well into this journal. The results are summarized nicely, and the discussion does well highlighting caveats and uncertainty. My only major comment is around more clarification needed for some of the methods. Otherwise, mostly minor comments. I'm giving it a minor revision as I don't think the "major comment" will take all that much work.

Thank you for the review of our manuscript. We greatly appreciate your constructive comments and suggestions. Please find our point-by-point response below (in blue). The line numbers in our responses are related to the revised version.

**Major comments**

Several aspects of the data and methods around the HBV model were unclear and could be improved by adding more details. One concern I have that I don't think was addressed was using station-based data as model inputs for the Czech catchments but then using gridded data for the Swiss catchments. Can the authors comment on what impact the station-based vs. gridded inputs might have on the simulations? Also, what was the spatial resolution of the gridded data used? It also was not clear to me what the HBV model inputs were. The historical dataset consists of temperature, precipitation, runoff, and SWE. Were all of these input into the HBV model?

Thank you for this relevant comment. Indeed, different types of input model data (gridded vs. stational) have been used for Czech and Swiss catchments. According to the results presented in Fig. 2, the model performs slightly better for Swiss catchments, where the gridded data has been used. This implies the model should perform better for Czech catchments if gridded data is used. However, only stational data were available at the time of study elaboration. Nowadays, official gridded products are available in Czechia as well, which would potentially enable the model calibrations to be redone using these input data. Although we are planning to do so, this is instead a long-term run and, unfortunately, can't be done for this study.

Nevertheless, we previously used model simulations using stational data in some of our recent studies (e.g., Jenicek and Ledvinka, 2020; Nedelcev and Jenicek, 2021; Hotovy et al., 2023) and did not find any major inconsistencies or errors in the resulting simulations. In the above studies, we also did several tests of the model performance, including its ability to simulate SWE (Nedelcev and Jenicek, 2021) or rain-on-snow events (Hotovy et al., 2023) correctly. These above tests also showed that although the absolute values of runoff signatures may change when different forcing data are used, we expect that the relative differences between individual perturbations (as the major approach used in our study) remain similar for different calibration runs. Thus they will not affect the interpretation of the overall results. Additionally, based on the reviewer 1 suggestion we tested how selected model parameters (threshold temperature $T_T$ differentiating between snow and rain and SWE threshold used to define RoS days) influence the number of defined RoS days (see the responses to the respective Reviewer 1 comments). The results of these new tests are discussed on lines 412-418 and 448-452 of the revised manuscript. For discussion related to previous model testing studies, we refer to L 381-398.

Additionally, we provided information about the spatial resolution of gridded data (2km for air temperature and precipitation, and 1km for SWE) and better described the HBV model inputs (see the method section, L 116-120 and L 124-127).

**Other comments**

Figure 1: I think this could be improved. As is, the polygons are very small and hard to see. I might suggest and panel plot with several zoomed maps that also include more detailed terrain.

We agree that it is not easy to find the best way to visualize such data. Based on your comment, we prepared several alternatives to the figure. We decided on one with two different maps separately for Czechia and Switzerland combined in one figure, with more detailed terrain as suggested (see new Fig. 1).

I would suggest using the term "perturbations" instead of "projections" throughout the paper. Projections are usually associated with GCM outputs in the future, and the future periods are not directly accounted for in your method.

Thank you for your suggestion. Although we have discussed earlier what terminology should be used here, we agree with your suggestion and changed the term to "perturbation" across the whole manuscript and all figures.

Line 190: What is the median objective function? Is a value of 1 a perfect score? Please explain.

Yes, a value of 1 represents a perfect fit. We slightly reworded the sentence to be clearer (L 205).

Figure 3: Great figure! Is the annual number of ROS days the mean? Median? Please specify in the caption.

Thank you. The annual number of RoS represents the annual mean from 1980-2010. We specified the figure caption in this respect to avoid confusion.

Figure 7: It looks like the T0_P12 line is drawn wrong in 7b. It shows a solid line and based on the text in lines 276-277 it should be small-dashed.

Thank you for double-checking the figure. Indeed, this was a mistake which we corrected in the revised manuscript. We slightly modified the figure's appearance as well. (see new Fig. 7)

**References**

Hotovy, O., Nedelcev, O., & Jenicek, M. (2023). Changes in rain-on-snow events in mountain catchments in the rain–snow transition zone. *Hydrological Sciences Journal*, *68*(4), 572–584. https://doi.org/10.1080/02626667.2023.2177544

Jenicek, M., & Ledvinka, O. (2020). Importance of snowmelt contribution to seasonal runoff and summer low flows in Czechia. *Hydrology and Earth System Sciences*, *24*(7), 3475–3491. https://doi.org/10.5194/hess-24-3475-2020

Nedelcev, O., & Jenicek, M. (2021). Trends in seasonal snowpack and their relation to climate variables in mountain catchments in Czechia. *Hydrological Sciences Journal*, *66*(16), 2340–2356. https://doi.org/10.1080/02626667.2021.1990298

---

## Referee Report (RR1)

The authors have done an excellent job responding to my comments and to the other reviewer. I have just found a few minor technical corrections that should be addressed before acceptance.

Comments:

Lines 69-80: There are at least three places (line 69, line 72, line 80) where the word "is" is used and the word "are" should be used instead. Please check the rest of the manuscript for similar grammar errors.

Figure 1: Nice job on remaking this figure! Can you add to the caption what "CH" and "CZ" mean? The figure and caption should be stand alone and the reader should not have to go back to the text to find these abbreviations.

---

## Author Response (AR2)

**Authors' response**

Dear Editor,

We would like to thank you and the reviewer for handling the manuscript and valuable comments. We have prepared a revised version of the manuscript that takes into account the reviewer's comments.

On behalf of all co-authors,

Michal Jeníček

**Review 2**

The authors have done an excellent job responding to my comments and to the other reviewer. I have just found a few minor technical corrections that should be addressed before acceptance.

Comments:

Lines 69-80: There are at least three places (line 69, line 72, line 80) where the word "is" is used and the word "are" should be used instead. Please check the rest of the manuscript for similar grammar errors.

We checked the mentioned lines, and we believe the word "is" should be used in all cases as this verb is related to nouns in their singular forms (a lack of, effect, understanding). However, we checked the manuscript for any other potential grammar errors or typos.

Figure 1: Nice job on remaking this figure! Can you add to the caption what "CH" and "CZ" mean? The figure and caption should be stand alone and the reader should not have to go back to the text to find these abbreviations.

We modified the legend caption in the figure to avoid abbreviations. Instead of abbreviations, the full names of both countries are presented in the figure, which is also consistent with the figure caption.